# Effects of univariate and multivariate bias correction on hydrological impact projections in alpine catchments

Judith Meyer[1a], Irene Kohn[1], Kerstin Stahl[1], Kirsti Hakala[2], Jan Seibert[2], and Alex J. Cannon[3]

[1]Faculty of Environment and Natural Resources, University of Freiburg, Freiburg, 79098, Germany
[2]Department of Geography, University of Zurich, Zurich, 8057, Switzerland
[3]Climate Research Division, Environment and Climate Change Canada, BC V8W 2Y2, Victoria, Canada

[a] now at: Catchment and Eco-Hydrology Research Group, Luxembourg Institute of Science and Technology, 4362, Esch-sur-Alzette, Luxembourg

*Correspondence to*: Irene Kohn (irene.kohn@hydrology.uni-freiburg.de)

**Abstract.** Alpine catchments show a high sensitivity to climate variation as they include the elevation range of the snow line. Therefore, the correct representation of climate variables and their interdependence is crucial when describing or predicting hydrological processes. When using climate model simulations in hydrological impact studies, forcing meteorological data are usually downscaled and bias corrected, most often by univariate approaches such as quantile mapping of individual variables neglecting the relationships that exist between climate variables. In this study we test the hypothesis that the explicit consideration of the relation between air temperature and precipitation will affect hydrological impact modelling in a snow-dominated mountain environment. Glacio-hydrological simulations were performed for two partly glacierized alpine catchments using a recently developed multivariate bias correction method to post-process EURO-CORDEX regional climate model outputs between 1976 and 2099. These simulations were compared to those obtained by using the common univariate quantile mapping for bias correction. As both methods correct each climate variable's distribution in the same way, the marginal distributions of the individual variables show no differences. Yet, regarding the interdependence of precipitation and air temperature, clear differences are notable in the studied catchments. Simultaneous correction based on the multivariate approach lead to more precipitation below air temperatures of 0 °C and therefore more simulated snowfall than with the data of the univariate approach. This difference translated to considerable consequences for the hydrological responses of the catchments. The multivariate bias correction forced simulations showed distinctly different results for projected snow cover characteristics, snowmelt-driven streamflow components, and expected glacier disappearance dates. In all aspects – the fraction of precipitation above and below 0 °C, the simulated snow water equivalents, glacier volumes, and the streamflow regime – simulations resulting from the multivariate-corrected data corresponded better with reference data than the results of univariate bias correction. Differences in simulated total streamflow due to the different bias correction approaches may be considered negligible given the generally large spread of the projections, but systematic differences in the seasonally delayed streamflow components from snowmelt in particular will matter from a planning perspective. While this study does not allow concluding definitively that multivariate bias

correction approaches are generally preferable, it clearly demonstrates that incorporating or ignoring inter-variable relationships between air temperature and precipitation data can impact the conclusions drawn in hydrological climate change impact studies in snow-dominated environments.

## 1 Introduction

With global change, hydrological processes in high elevation regions have been significantly impacted (Messerli et al., 2004). In the European Alps, the observed increase in air temperature is a trend that is expected to continue in the future. Future precipitation changes are less clear, with an expected slight increase in winter precipitation (Gobiet et al., 2014; Kotlarski et al., 2016). The hydrology of alpine catchments is especially sensitive to these changing climate variables (Köplin et al., 2010). High elevations in the Alps are still characterized by snow cover and the existence of glaciers.
However, rising air temperatures and a consequent upward shift of the zero-degree isotherm has led to a decrease in snow accumulation and an increase in glacier melt (Pellicciotti et al., 2010). Due to shrinking glacier areas, the glacial influence in the streamflow regimes has decreased. This is especially notable during late summer when water from ice melt can constitute a notable percentage of total streamflow. With progressive glacier retreat, the ice melt contribution to streamflow is expected to decrease (Jansson et al., 2003; Hock, 2005; Moore et al., 2009; Huss and Hock, 2018). The interdependence of air
temperature and precipitation is particularly important for hydrological systems as it determines the physical state of precipitation. Bosshard et al. (2014) showed that an air temperature dependent shift from snowfall to rain has notable effects on catchment water storage and seasonal water availability in such an environment. A correct representation of climate variables and their interdependence is therefore essential in hydrological simulations of glacierized catchments.

In hydrological climate change impact studies, post-processing of climate model data has become a standard procedure.
Despite continuous progress, raw outputs from regional climate models differ largely from observational reference data due to both spatial mismatches and systematic biases. Therefore, climate model outputs are downscaled and biases are adjusted statistically before being used in hydrological simulations (Ehret et al., 2012; Maraun, 2016; Teutschbein and Seibert, 2012). Many empirical statistical techniques have been developed to post-process climate model outputs for these purposes. For hydrological impact studies quantile mapping approaches, which correct for biases in the data's entire distribution, have
often been recommended (Teutschbein and Seibert, 2012; Gudmundsson et al., 2012; Chen et al., 2013). However, these approaches correct the climate variables independently from one another. The interdependence of key climate variables, such as air temperature and precipitation, can be especially important when modelling snow-dominated catchments due to the aforementioned threshold effects of the transition of rain to snowfall or the conditions required for snow and ice melt.

Studies that analyzed inter-variable aspects of bias correction showed that univariate quantile mapping retains the inter-
variable dependencies as represented by the raw climate model output data (Wilcke et al., 2013; Ivanov and Kotlarski, 2017). But, these may not correspond to the local interdependencies in observations. To account for interdependencies, multivariate bias correction approaches have been developed that allow for the preservation of the interdependence of climate variables as

represented by the target observation data throughout the bias correction process (Li et al., 2014; Cannon, 2016, 2018; Mehrotra and Sharma, 2016, 2015). A correction procedure that preserves the climate variables' interdependence may be considered more appropriate for subsequent impact analyses, such as the application of a calibrated hydrological model using multiple variables, than univariate techniques that ignore biases in inter-variable relationships (Cannon, 2018).

While many studies have evaluated bias correction methods in terms of their effects on the actual variables of precipitation and air temperature themselves, studies that use impact models to investigate the consequence of bias correction in the modelled impacts are still rare. So far, there have been only a few studies (Räty et al., 2018; Chen et al., 2018) that investigated the effect of using a multivariate bias correction technique on hydrological projections. Chen et al. (2018) found that jointly corrected precipitation and air temperature data better modelled eleven out of twelve catchments in the

calibration period than the meteorological data that was corrected with a univariate method. An advantage of using a bivariate bias correction approach was not evident for the coldest snow-dominated catchment of the sample though. Hydrological simulations by Räty et al. (2018) generally did not substantially benefit from bivariate bias correction approaches, but when looking more specifically, simulations of high flows and snow water equivalents in snow-influenced catchments improved slightly.

In this study we investigate the hypothesis that the explicit consideration of the relation between air temperature and precipitation in bias correction will affect hydrological impact modelling in snow- and glacier melt dominated environments. Here, dependencies are known to matter most as they have cumulative effects over a season through snow storage and at multi-year time scales through the glacier mass balance. The approach of this study was therefore to conduct climate impact modelling experiments that allow comparison of the effects of univariate and multivariate bias correction of precipitation and

air temperature input on the hydrological change in alpine catchments. The model experiments were conducted for two meso-scale partly glacierized catchments in the Swiss Alps, for which snow accumulation, glacier mass balance, and streamflow were simulated from 1976 to 2099.

## 2 Study catchments and data

### 2.1 Study area

Two partly glacierized meso-scale catchments in the Swiss Alps, in the headwater of the Rhine River, were examined in this study: the Hinterrhein catchment and the larger Schwarze Lütschine catchment (Fig. 1, Table 1). Based on the dataset by Freudiger et al. (2018), used in this study, around the year 1900 glacier coverage was approximately 32% of the Hinterrhein catchment area and around 25% of the Schwarze Lütschine catchment area. Glaciers in both catchments retreated considerably during the 20[th] century. The Hinterrhein catchment is characterized by small, scattered glaciers, which by 1973

lost around half their area, leading to a glacier coverage of only 7% in 2010 (Table 1). In the Schwarze Lütschine catchment

losses in relative glacier area have been smaller. This difference in glacier coverage is related to elevation with considerably higher maximum elevations in the Schwarze Lütschine catchment compared to the Hinterrhein catchment (Table 1).

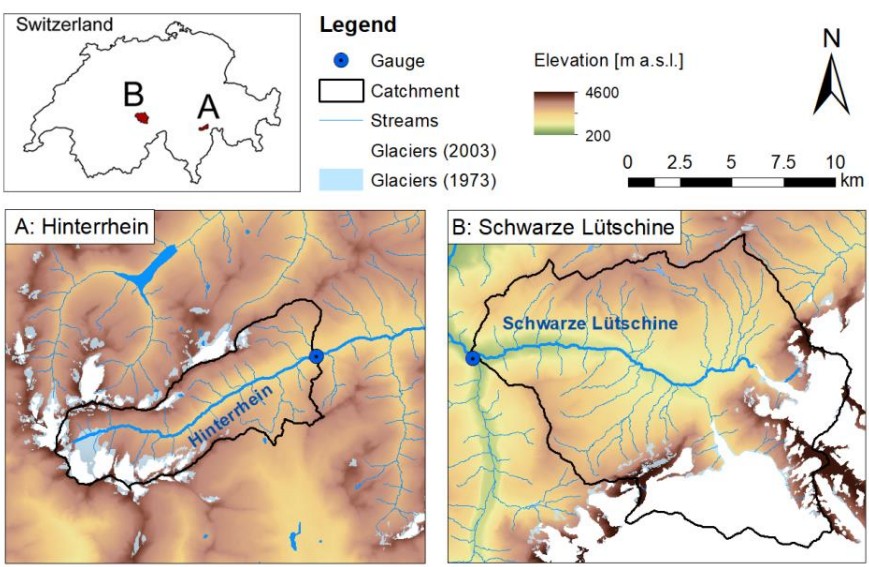

Figure 1: Map of the two study catchments and their location in Switzerland: Hinterrhein (A) and Schwarze Lütschine (B).

Table 1: Catchment characteristics including glacier cover information.

| | Area | Elevation | | | Glacier cover[*] | | | | | |
| | | mean | min | max | 1973 | | 2003 | | 2010 | |
| | [km²] | [m a.s.l.] | | | [km²] | [%] | [km²] | [%] | [km²] | [%] |
|---|---|---|---|---|---|---|---|---|---|---|
| Hinterrhein | 53.9 | 2357 | 1587 | 3387 | 9.1 | 17.8 | 4.7 | 8.7 | 3.8 | 7.1 |
| Schwarze Lütschine | 179.9 | 2059 | 648 | 4086 | 37.0 | 23.5 | 34.4 | 19.1 | 29.7 | 16.5 |

[*] Based on glacier inventories by Müller et al. (1976) / Maisch et al. (2000) for 1973, Paul et al. (2011) for 2003, and Fischer et al.(2014) for 2010.

## 2.2 Data and data preparation

The application of bias correction algorithms to climate model outputs is generally based on three datasets: historical observations as reference (also called 'target') data, historical climate model simulations, and the corresponding climate model projections. In the present study the historical reference data for the study catchments were derived from an observation based interpolation product, i.e. the 1x1 km² gridded daily air temperature and precipitation datasets from the HYRAS product (Rauthe et al., 2013; Frick et al., 2014). Area-weighted mean values of precipitation and air temperature

were extracted for the study catchments. The extracted catchment mean precipitation time series were corrected for undercatch based on the method by Sevruk (1989) and were then further adjusted through validation with long-term annual mean precipitation sums resulting from a water balance approach (for details see Stahl et al., 2017). The resulting time series of catchment mean precipitation and air temperature were used as input for the calibration of the glacio-hydrological model

and as historically observed climate data (HOCD) for the bias correction.

The climate model datasets were obtained from the Coordinated Regional Climate Downscaling Experiment (CORDEX, www.cordex.org) via the Earth System Grid Federation (ESGF) archive (http://www.cordex.org/data-access/esgf/). CORDEX is a collaborative effort within the climate modelling community where general circulation models (GCMs) are downscaled using regional climate models (RCMs). Since all catchments in this study are located in Switzerland, GCM–

RCMs were selected from the European domain of the CORDEX project (EURO-CORDEX, http://www.euro-cordex.net/). EURO-CORDEX provides simulations at 0.11° (~12.5 km horizontal resolution) and 0.44° (~50 km horizontal resolution). Given that the catchments used in this study are situated in the Alpine domain, only the higher resolution 0.11° simulations were used. Two Representative Concentration Pathways (RCPs) were selected for this study: RCP 4.5 represents an intermediate mitigation scenario, where greenhouse gas (GHG) emissions will peak around 2040 and then steadily decrease,

and RCP 8.5 represents a more pessimistic scenario, which assumes that GHG emissions will continue to increase throughout the 21$^{st}$ century (Meinshausen et al., 2011).

Precipitation (P) and air temperature ($T_a$) data were provided by the ten GCM–RCMs shown in Table 2 for the time period 1970–2099. For each catchment, raw GCM–RCM data were extracted using an area-weighted method as shown in Hakala et al. (2018). Based on the areal fraction of an RCM grid cell overlying a particular catchment, five RCM grid cells contribute

to each catchment. All GCM–RCMs used in this study utilize a Gregorian calendar.

**Table 2: GCM–RCM combinations from the EURO-CORDEX initiative used in this study.**

| Driving GCM | RCM | RCM institution |
|---|---|---|
| CNRM-CM5-LR [1] | CCLM4-8-17 | Climate Limited-area Modelling Community |
| CNRM-CM5 [1] | RCA4 | Swedish Meteorological and Hydrological Institute |
| EC-EARTH [2] | CCLM4-8-17 | Climate Limited-area Modelling Community |
| EC-EARTH [2] | HIRHAM5 [5] | Danish Meteorological Institute |
| EC-EARTH [2] | RACMO22E [5] | Royal Netherlands Meteorological Institute |
| EC-EARTH [2] | RCA4 | Swedish Meteorological and Hydrological Institute |
| IPSL-CM5A-MR [3] | WRF331F | Laboratoire des Sciences du Climat et de l'Environnement |
| IPSL-CM5A-MR [3] | RCA4 | Swedish Meteorological and Hydrological Institute |
| MPI-ESM-LR [4] | CCLM4-8-17 | Climate Limited-area Modelling Community |
| MPI-ESM-LR [4] | RCA4 | Swedish Meteorological & Hydrological Institute |

GCM institutions:   [1] CNRM-CERFACS (Centre National de Recherches Météorologiques-Centre Européen de Recherche et de Formation Avancée en Calcul Scientifique); note that a warning concerning an inconsistency in the historical run of CNRM-CM5 has been issued on the CORDEX errata page ( https://www.euro-cordex.net/078730/index.php.en) after data had been downloaded and selected for this study, [2] EC-Earth consortium, [3] IPSL (Institut Pierre-Simon Laplace), [4] MPI-M (Max Planck Institute for Meteorology)

[5]   CORDEX errata page ( https://www.euro-cordex.net/078730/index.php.en) notes snow accumulation issues for these RCM runs.

The application of the hydrological model requires catchment mean time series of $P$ and $T_a$. These were subjected to bias correction. Further data used as model input and for model calibration were not directly bias corrected. Daily potential evapotranspiration was calculated with an air temperature based approach provided by Oudin et al. (2005). Catchment specific air temperature lapse rates were determined based on daily values from the HYRAS product. Based on the reference period from 1976–2006 a mean for each day of the year was calculated and smoothed using an 11-day moving average. A mean precipitation gradient (in % per 100 m a.s.l.) was determined from the corrected HYRAS data and applied as constant value in all simulations.

Daily streamflow data for model calibration were provided by the Swiss Federal Office for the Environment (FOEN) and the "Amt für Wasser und Abfall des Kantons Bern". The available streamflow record for the station Gündlischwand (operated by the Cantone of Berne) at the outlet of the Schwarze Lütschine study catchment covered only the period 1992–1999. By using the record of a downstream station of the Lütschine River (station Gsteig) and subtracting the streamflow of its other major headwater tributary (record from the station Zweilütschinen of the Weisse Lütschine) the streamflow for the Schwarze Lütschine study catchment could be reconstructed for the entire simulation period. This reconstructed streamflow time series was validated with the available streamflow data from the station Gündlischwand for the subperiod 1992–1999 and then used for model calibration. Snow water equivalent (SWE) and snow cover data were derived from a snow map (interpolated grid) product by the OSHD-SLF (2013). The glacier area was assessed based on glacier inventory data by Müller et al. (1976) and Maisch et al. (2000) for the state in the year 1973, by Paul et al. (2011) for the state in 2003, and by Fischer et al. (2014) for the year 2010 (see Table 1). Estimates of glacier volume were derived based on gridded ice thickness data available for the years 1973 and 2010, which were computed using the approach by Huss and Farinotti (2012) and provided by Matthias Huss. Glacier volume for the year 2003 was estimated based on the glacier cover according to Paul et al. (2011) and glacier volume–area scaling. The glacier volume estimate for 1973 was used for model initialization. The estimate for 2003 was incorporated in the model calibration for the period 1976–2006. The estimate for 2010 was not directly used in the calibration but served the validation of model simulations beyond the year 2006.

## 3 Methods

### 3.1 Bias correction of climate data

Depending on the GCM–RCM combination, raw climate variables (noBC) of the control period (1976–2006) differ from the reference data (HOCD). To correct these biases, two different bias correction methods were applied to each climate model's $T_a$ and $P$ series: a univariate quantile mapping technique – Quantile Delta Mapping (QDM) – and a multivariate bias correction approach (MBCn). Quantile mapping is based on a transfer function that transforms the cumulative distribution(s) of the modelled data to match the distribution(s) of the observed series. The obtained transfer function is then applied to all climate model data, historical and projected. Thus it corrects systematic distributional biases relative to historical observations and preserves model-projected relative changes. Quantile Delta Mapping (QDM) is a variant of quantile

mapping by Cannon et al. (2015) that was designed to avoid artificial deterioration of trends arising as a statistical artefact of standard quantile mapping. QDM corrects systematic distributional biases relative to historical observations and preserves model-projected changes in quantiles in the projection period. For a given time slice, the climate model's change signal ($\Delta$) – relative change for precipitation and absolute change for air temperature – is removed from all projected future quantiles in a first step. Quantile mapping is then applied before the projected changes in quantiles are reintroduced to the bias corrected model output.

The MBCn multivariate bias correction algorithm by Cannon (2018) is based on the N-dimensional probability density function transform. This approach was originally developed for image processing (Pitié et al., 2007) but has been converted for post-processing climate model data. MBCn combines QDM and random orthogonal rotations to match the multivariate distributions of climate model data and observed data. In the MBCn approach, a random orthogonal rotation of the data points is applied before QDM. This exposes QDM to a linear combination of the original variables, which is then used to correct the marginal distributions of the rotated data. The QDM-corrected dataset is then rotated back and convergence to the observed multivariate distribution is checked. These steps are conducted iteratively until the multivariate distributions of bias corrected climate model data and observed climate data match. In this study, 100 iterations were conducted. Both QDM and MBCn were applied in a seasonally dependent fashion. Specifically, bias corrections were applied over 30-year sliding windows. This involved replacing the central 10-years and sliding forward 10-years for each 30-yr window, until the end of the projection period was reached. Within each window – to ensure an unbiased seasonal cycle – bias corrections were applied separately for each calendar month. The combination of change-preservation by QDM, which is also a core component of MBCn, with sliding windows ensures that projected trends from the underlying climate model are largely preserved. This follows the general approach and recommendation of Hempel et al. (2013) concerning trend preservation of post-processed climate model output for impact modelling.

Climate model data is often simultaneously bias corrected and downscaled as the reference data stems from stations or higher resolution observations in comparison to the coarse grid resolution of RCMs. Undesirable effects in downscaling to finer scales have been one of the major limitations of current bias correction methods (Maraun, 2013; Ehret et al., 2012; Maraun et al., 2017). Such artefacts can occur especially in complex terrain and if the scale gap between climate model outputs and impact model data is considerable. In general, bias correction based on spatial resolutions that differ substantially should be avoided or handled with great care. In this study the discrepancy in resolution is assumed acceptable as the bias correction was based on spatially aggregated mean climate variables for the meso-scale catchments (54 km² and 180 km²) with the original resolution of the underlying gridded datasets (GCM–RCM data: 0.11°, historical HYRAS data: 1 km) becoming of secondary importance.

**3.2 Hydrological model simulations**

The HBV model (Bergström, 1976; Lindström et al., 1997) is a semi-distributed bucket-type runoff model. Here the software implementation HBV-light (Seibert and Vis, 2012) was used, which recently has been extended to represent coupled glacio-

hydrological processes of partly glacierized catchments (Seibert et al., 2018). This version of the HBV model also allows tracking the different components of streamflow resulting from rainfall ($Q_R$), snowmelt ($Q_S$), and glacier ice melt ($Q_I$) (Weiler et al., 2018; Seibert et al., 2018). The HBV model requires daily precipitation, air temperature, and potential evapotranspiration data as input to simulate daily runoff. In addition, linear gradients of air temperature and precipitation are

needed for the interpolation over elevation zones. A general description of the basic model structure and the process conceptualization of the HBV model are found elsewhere (e.g., Lindström et al., 1997; Seibert and Vis, 2012; Seibert et al., 2018). Snow and ice accumulation and melt are based on a widely used air temperature index approach using a threshold air temperature as a model parameter to differentiate between precipitation falling as snow and rain as well as to simulate melt of snow and ice by additionally using a degree-day factor. Differences in the melt of glacier ice compared to snow are

represented by another model parameter. The influence of differences in aspect on snow and ice melt was taken into account by distinguishing three aspect classes and applying an additional aspect factor parameter (Hagg et al., 2007; Hottelet et al., 1993). The latest version of the HBV-light software with the implementation of the coupled glacio-hydrological processes and the adjustment of glacier geometry to glacier mass changes based on the *Δh*-parametrization by Huss et al. (2010) is explained in detail in Seibert et al. (2018). It should be noted that with the implementation in HBV-light only one glacier per

catchment or subcatchment can be represented. Hence, glacier cover areas in each of the two case study catchments were aggregated and simulated as one 'virtual' model glacier.

The model was calibrated for the period from 1976–2003, preceded by a 3-year warm-up period, by optimizing a weighted objective function, giving special attention to streamflow dynamics (50%), snow simulation (25%), and glacier volume change (25%). The Lindström measure (Lindström, 1997) was used for the streamflow's general dynamic and volume errors,

while the Nash–Sutcliffe efficiency (Nash and Sutcliffe, 1970) was computed based on logarithmically-transformed streamflow. Additionally the Nash–Sutcliffe efficiency was computed for the streamflow only during the summer months from June to September. To calibrate the snow simulations the snow covered area fraction of the catchment as well as the mean SWE of the elevation range < 2500 m a.s.l. were used. Elevations below 2500 m a.s.l. represent the crucial range for the snow line and in this range the gridded SWE interpolation used as reference data is well-founded on station data. Glacier

volume was considered in the calibration process using glacier volume estimates for the years 1973 and 2003. The automated multi-criteria calibration was based on a genetic algorithm for parameter optimization (see Seibert, 2000). A 3-year model validation period (2003/10/01–2006/12/31) completed the historical reference period 1977–2006. Resulting performance measures for the calibration and validation period are summarized in Table 3 (see Supplement for additional figures comparing simulated variables and reference data). The retreat of the glaciers required all experiments to be run in a

transient mode, i.e. the model was forced with climate model scenario data for the period from October 1976 to September 2099.

**Table 3: Model performance criteria for the calibration (1976/10/01–2003/09/30) and validation (2003/10/01–2006/12/31) of the hydrological model formulated (see footers) that the ideal value for a perfect fit is 1.0.**

| Model performance criteria | Weight in calibration | Hinterrhein | | Schwarze Lütschine | |
|---|---|---|---|---|---|
| | | Calibration | Validation | Calibration | Validation |
| Nash–Sutcliffe efficiency ($R_{eff}$) [5] for streamflow | - | 0.773 | 0.763 | 0.910 | 0.880 |
| Kling–Gupta efficiency [6] for streamflow | - | 0.861 | 0.877 | 0.934 | 0.898 |
| Volume error ($V$) [7] for streamflow | - | 0.972 | 0.962 | 1.000 | 0.965 |
| Lindström measure [8] for streamflow | 0.20 | 0.770 | 0.759 | 0.910 | 0.877 |
| $R_{eff}$ [5] for log transformed streamflow | 0.15 | 0.840 | 0.648 | 0.908 | 0.749 |
| $R_{eff}$ [5] for streamflow in Jun–Sep | 0.15 | 0.684 | 0.711 | 0.795 | 0.749 |
| Root mean square error for snow covered area fraction [9] | 0.10 | 0.856 | 0.761 | 0.863 | 0.803 |
| Mean absolute normalized error (MANE) for SWE [10] | 0.20 | 0.642 | 0.557 | 0.757 | 0.553 |
| Glacier volume change objective function [11] | 0.20 | 0.999998 | - | 0.999994 | - |

Formulation of model performance criteria:

[5] $R_{eff} = 1 - \frac{\Sigma(Q_{obs}-Q_{sim})^2}{(Q_{obs}-\overline{Q_{obs}})^2}$ where $Q_{obs}$ and $Q_{sim}$, respectively, are observed and simulated streamflow [mm/day]

[6] see Gupta et al. (2009)

[7] $1 - V$ with $V = \frac{\Sigma|(Q_{obs}-Q_{sim})|}{\Sigma(Q_{obs})}$ where $Q_{obs}$ and $Q_{sim}$, respectively, are observed and simulated streamflow [mm/day]

[8] $1 - R_{eff} - 0.1\,V$ with $R_{eff}$ [5] and $V$ [6] see Lindström et al. (1997)

[9] $1 - \sqrt{\frac{1}{n}(C_{sim} - C_{ref})^2}$ with $C_{ref}$: snow covered catchment area fraction ($C$) [-] as per gridded SWE reference data; $C_{sim}$: simulated $C$; $n$: number of time steps

[10] $1 - \frac{\Sigma|(S_{ref}-S_{sim})|}{\Sigma S_{obs}}$ with $S$ [mm]: mean SWE for elevation range below 2500 m a.s.l. where $S_{ref}$ is derived from SWE reference data and $S_{sim}$ is simulated

[11] $1 - \frac{|\Delta W_{sim}-\Delta W_{obs}|}{\Delta W_{obs}}$ with $\Delta W$ [mm]: change of glacier ice volume in water equivalent between the years 1973 and 2003, where $\Delta W_{obs}$ corresponds to an estimate based on observed glacier area and $\Delta W_{sim}$ is simulated

## 3.3 Data analysis

5   Effects of the bias correction approaches on the hydrological simulation were based on comparisons of the simulation results for the historical reference period 1976–2006 using $P$ and $T_a$ time series derived from the HYRAS datasets as input (Sim$_{HOCD}$) and simulations forced with $P$ and $T_a$ series from the output of the ten different GCM–RCMs for the two different RCP scenarios, each uncorrected (Sim$_{noBC}$) and bias corrected based on QDM (Sim$_{QDM}$) and on MBCn (Sim$_{MBCn}$). In total, this led to 61 hydrological model runs (1 Sim$_{HOCD}$, 20 Sim$_{noBC}$, 20 Sim$_{QDM}$, and 20 Sim$_{MBCn}$) per catchment. In a first step

10  (Results Section 4.1), the different $P$ and $T_a$ series were evaluated for the amount of precipitation occurring at air temperatures above and below of 0 °C due to the importance for the simulation of snow accumulation and melt processes. Furthermore, the simulation results were assessed in terms of SWE, glacier ice volume ($V_I$) evolution (Results Section 4.2), and eventually streamflow with its three individual components $Q_R$, $Q_S$, and $Q_I$ (Results Section 4.3).

# 4 Results

## 4.1 Climate variables bias correction

The two applied bias correction methods led to differences concerning the interdependence of $P$ and $T_a$. The distribution of annual precipitation sums during air temperatures above and below 0 °C of the entire ensemble is represented in Fig. 2, while

5    results for the individual GCM–RCM output series are provided in the Supplement. Generally, the uncorrected climate model data (noBC) have a wider variability than the reference data (HOCD). Particularly for the Schwarze Lütschine the uncorrected data yielded precipitation amounts remarkably higher than historically observed. However, differences also existed between the correction methods. For both catchments precipitation falling above air temperatures of 0 °C was overestimated with QDM. Accordingly, precipitation falling below air temperatures of 0 °C was underestimated in the

10   univariate bias corrected data. MBCn appears to have better reproduced the historical reference data in this respect.

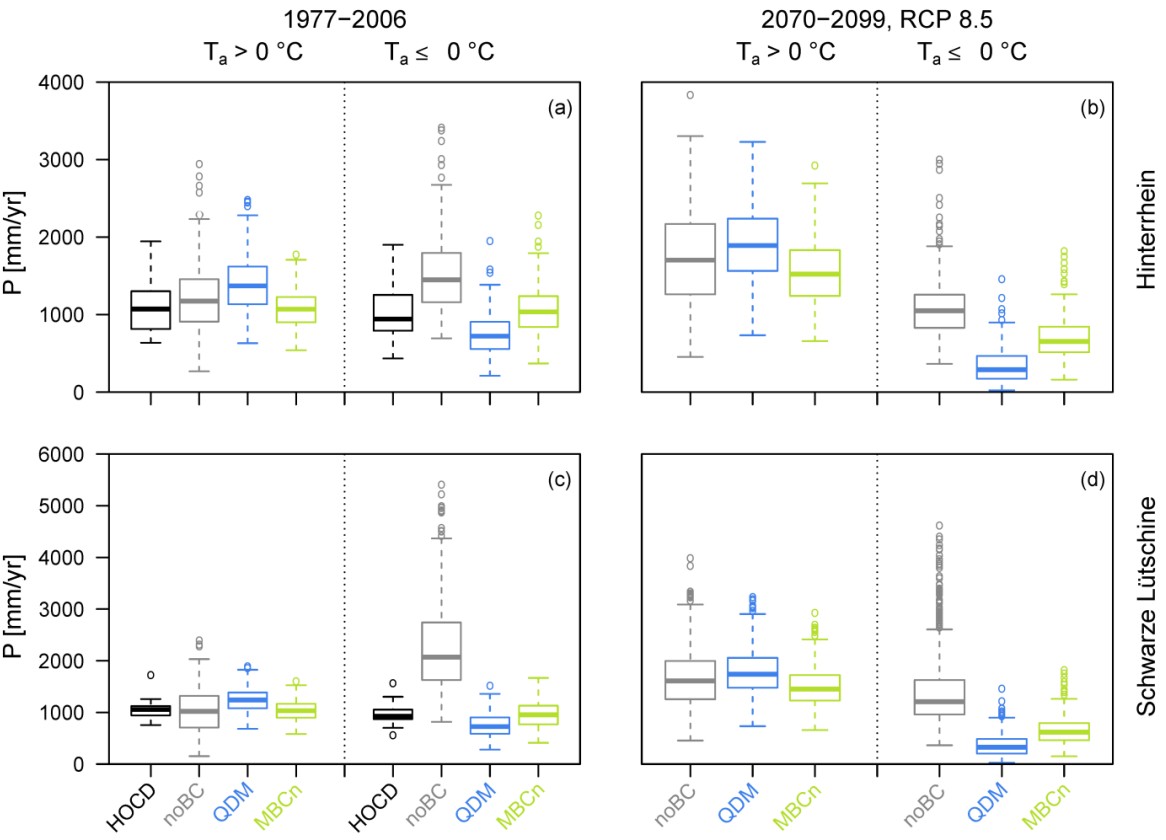

**Figure 2: Annual precipitation sums for days with air temperatures above or below 0 °C.**

## 4.2 Hydrological model simulations – cryosphere

Application of the climate scenarios clearly revealed a decreasing role of snow for both study catchments. Figure 3 illustrates a distinctly smaller snow accumulation in the course of a year simulated for the period 2070–2099 compared to the historical reference period (1977–2006) and a more complete melt during the summer. This extended the snow free period during the summer in the Hinterrhein catchment. The spread between the simulations diverged for the simulations of future conditions. In the Schwarze Lütschine catchment with its higher maximum elevations all effects were comparable, yet a permanent snow cover remained still present based on most scenarios. As expected, simulations based on the RCP 4.5 scenario (not shown) led to a clear but less severe decrease in mean SWE than for the RCP 8.5 scenario.

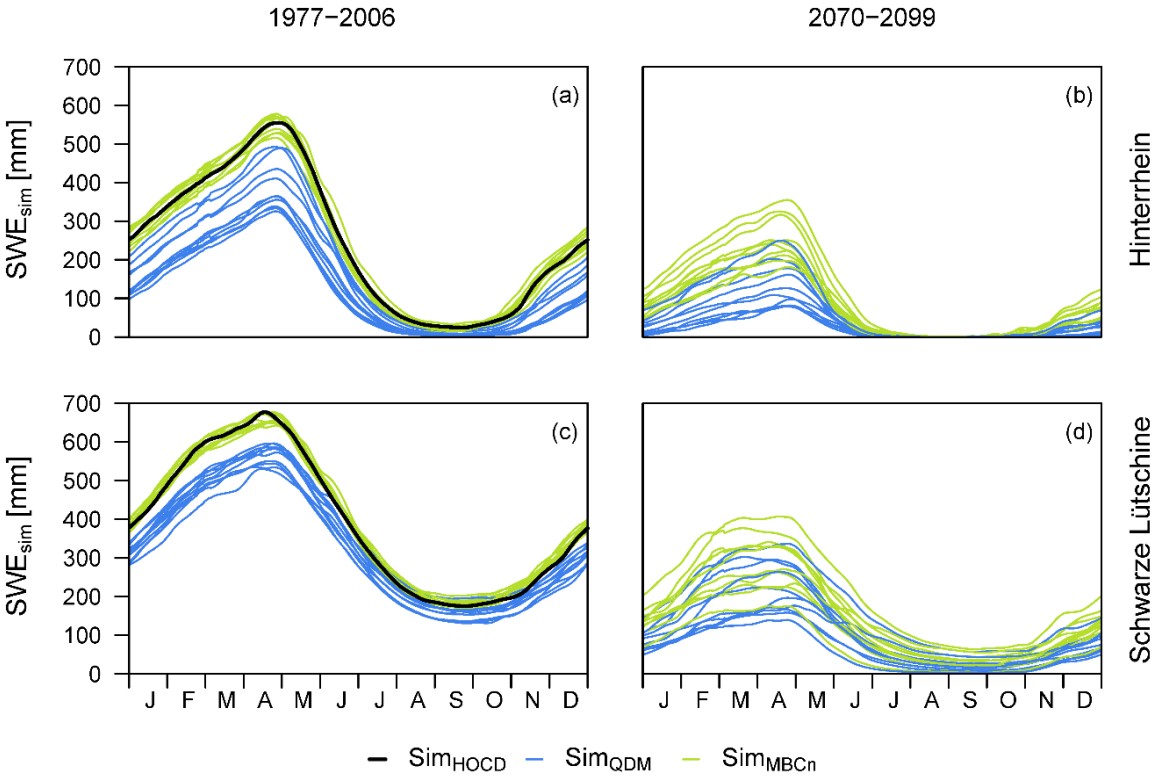

**Figure 3: Mean annual SWE regime, calculated using the 11-day moving average of daily simulated SWE (catchment mean) for a and c the historical reference period and b and d at the end of the scenario period based on the RCP 8.5 scenario.**

The differences in the interdependence of precipitation and air temperature resulting from the application of QDM versus MBCn to the GCM–RCM data can be seen in the simulated SWE (Fig. 3). The state of precipitation defined by the calibrated threshold air temperature parameter TT (Schwarze Lütschine TT = -0.29 °C; Hinterrhein TT = -0.73 °C) influenced the snow accumulation and therefore led to differences in the annual SWE regime (Fig. 3). As MBCn-corrected

GCM–RCM data caused more precipitation to fall as snow, the accumulated catchment mean SWE in spring was simulated to be up to around 100–200 mm higher in the historical reference period compared to simulations based on QDM-corrected forcing data. Simulated SWE based on the two different bias correction methods differed notably. Comparing the results with the reference simulation (Fig. 3) indicates that MBCn performed better. The systematic difference in simulated SWE resulting from the bias correction methods was a bit less clear for the Schwarze Lütschine catchment in the scenario period, yet overall the differing tendencies between QDM- and MBCn-corrected data were considerable.

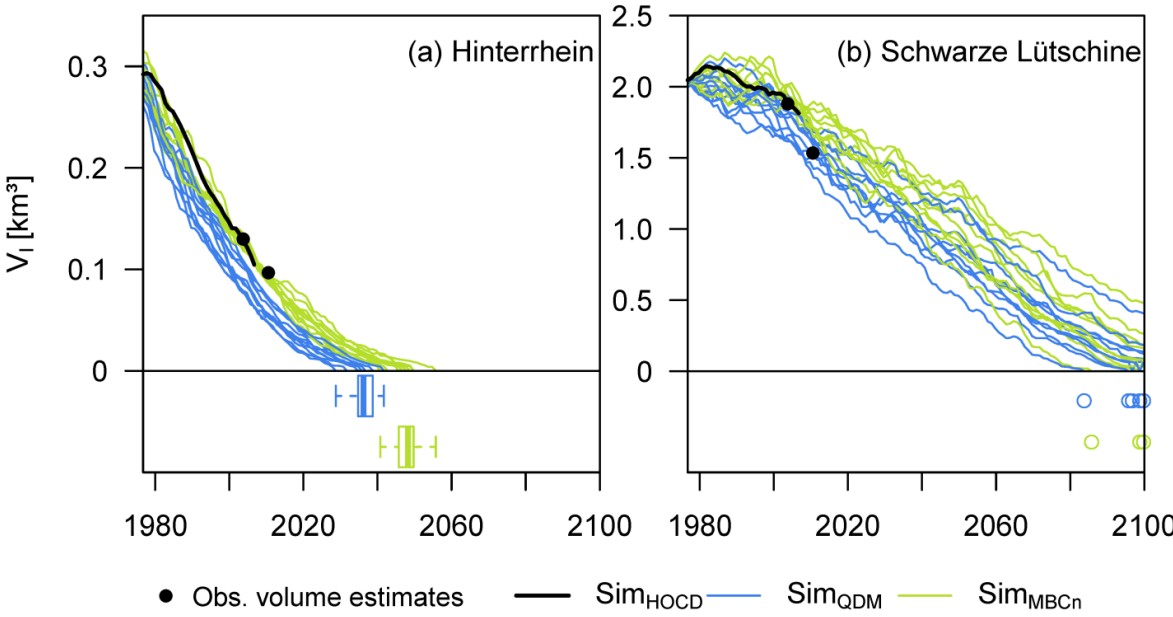

**Figure 4: Simulated glacier ice volume from 1977 to 2099 using the RCP 8.5 scenario forcing in the two catchments (a, b). In the lower part of the graphs the boxes in the left figure and the dots in both figures indicate the simulated years of the complete glacier ice melt. For the Schwarze Lütschine only 5 (3) out of the 10 Sim$_{QDM}$ (Sim$_{MBCn}$) simulations led to complete glacier melt by the end of 2099, not allowing to show any boxplots. Filled black circles are glacier volume estimates based on observed glacier area data in 2003 and 2010.**

For the period 1976 to 2099 the glacier volume was simulated to decrease in both catchments. In the Hinterrhein catchment, glaciers diminished continuously from the beginning of the simulation period and were simulated to have disappeared between 2028 and 2055 under the RCP 8.5 scenario depending on the GCM–RCMs and the applied bias correction method (Fig. 4). In the Schwarze Lütschine catchment, data from a few GCM–RCMs resulted in an increase in simulated glacier volume in the 1970s and 1980s, which is in line with the historical reference simulation (Sim$_{HOCD}$). In the following years, glacier volume decreased continuously. In contrast to the Hinterrhein catchment, glaciers were not simulated to have disappeared by the end of 2099 based on the RCP 4.5 scenario (not shown). However, in the simulations the glacier volume

diminished to on average roughly a third of its initial size at the beginning of the simulation period. The RCP 8.5 scenario from a few certain GCM–RCM combinations even led to complete glacier disappearance in the Schwarze Lütschine catchment within the 21[st] century.

Focusing on systematic differences between simulations using data corrected based on QDM and MBCn, the simulations of glacier volume showed similar tendencies as were found for SWE. For both catchments, but again more clearly for the Hinterrhein catchment, MBCn-corrected GCM–RCM data resulted in a slower decline in glacier volume in comparison to simulations based on QDM-corrected data. All projections led to complete glacier disappearance in the Hinterrhein catchment by about the year 2050 with a clear tendency towards earlier dates for QDM-based simulations (2028–2041, mean: 2036) compared to MBCn-based simulations (2040–2055, mean: 2047). For the Schwarze Lütschine catchment the range of QDM- and MBCn-based glacier volume simulations overlapped largely as simulations in general diverged considerably. However, for each individual GCM–RCM dataset, glacier melt was simulated to be faster using the QDM-corrected data compared to the MBCn-corrected data. The less intense decline in glacier volumes resulting from MBCn-corrected forcing data appeared to correspond better with the reference simulation ($Sim_{HOCD}$) in the initial phase of the historical period and with the observation-based glacier volume estimates for the year 2003 (and also for the year 2010 in case of the Hinterrhein catchment). MBCn thus led to more realistic results for the historical reference period.

### 4.3 Hydrological model simulations – streamflow

Time changes of annual variables and mean monthly hydrological regimes were assessed for streamflow $Q$ and for the individual streamflow components, i.e. the rain component $Q_R$, the snowmelt component $Q_S$, and the ice melt component $Q_I$. Mean annual streamflow of the study catchments showed a small decrease over the entire simulation period from 1976 to 2099 for most simulations, while for some a slight increase was noticed (Fig. 5). However, the simulations based on different GCM–RCM outputs diverged over time. While – on average – the total annual streamflow stayed largely unchanged, its composition changed clearly. The streamflow component from glacier ice melt decreased slowly over time as the glaciers retreated. Likewise, the snowmelt component of streamflow decreased over time. On average, for the RCP 4.5 scenario's MBCn-corrected data these decreases were around 14% in the Hinterrhein and 16% in the Schwarze Lütschine for the RCP 8.5 scenario's QDM-corrected data they were around 53% in the Hinterrhein and 33% in the Schwarze Lütschine.

The streamflow simulations reflected the changes from the different bias correction methods found for the cryosphere. Simulations based on QDM-corrected data led to slightly different total streamflow than MBCn-corrected data (Fig. 5 a, d, e). These differences were much more pronounced regarding the individual streamflow components. Modelling based on QDM-corrected climate data led to an approximately 10% higher rain component of streamflow $Q_R$ in comparison to MBCn-corrected simulations. The snowmelt component of streamflow $Q_S$ varies proportionally, being notably smaller when using QDM-corrected GCM–RCM data. Comparing the means of the ice melt components of streamflow $Q_I$ for the 30-year periods at the beginning and at the end of the entire simulation period showed no differences from the bias correction methods for the Hinterrhein catchment and differences in the range of only 1% for the Schwarze Lütschine catchment.

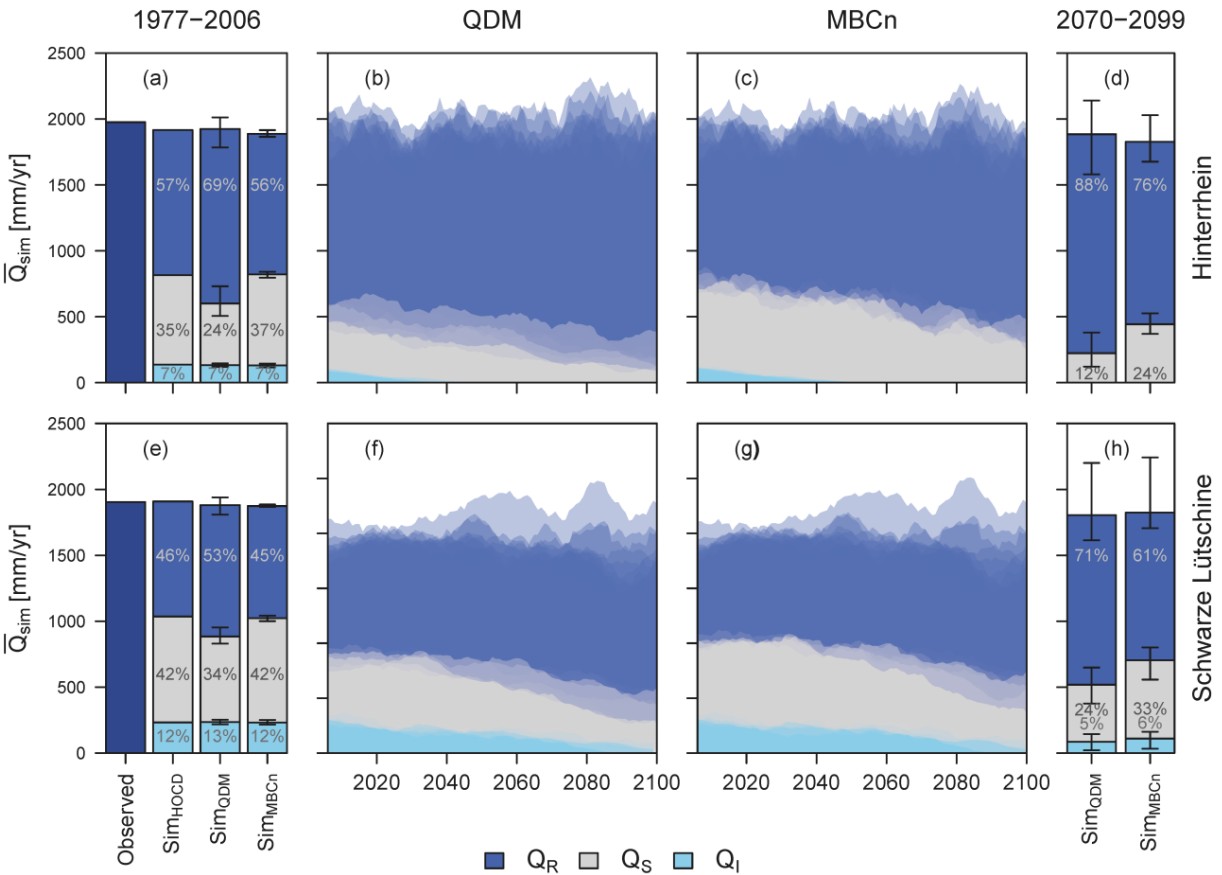

**Figure 5: Observed total streamflow and simulated streamflow components for the historical reference period and for the different simulations under the RCP 8.5 scenario. Stacked bar plots show mean values over the historical reference period (a, e) and for the period 2070–2099 (d, h), stacked bar plots for $Sim_{QDM}$ and $Sim_{MBCn}$ show ensemble mean with ensemble spread (error bars). Simulation results over the scenario period 2006–2099 (b, c, f, g) are shown as semi-transparent polygons for each GCM–RCM combination.**

Simulated streamflow and its components, $Q_I$, $Q_S$, and $Q_R$, also changed seasonally (Fig 6). In the historical reference period (1977–2006), the two catchments had a nivo-glacial streamflow regime peaking in the summer due to snow and ice melt and with little streamflow during winter. According to the projections the streamflow peak in early summer remained a dominant characteristic until the end of the simulation period. Yet, for the Hinterrhein catchment, the peak's timing was simulated to shift causing streamflow to concentrate in May and the peak to become much narrower than in the past. For the Schwarze Lütschine catchment the simulations for the RCP 8.5 scenario resulted in very variable summer streamflow regimes for 2070–2099 and a tendency towards a lower summer streamflow peak than in the past. In the reference period, the glaciers' influence showed during late summer, where it extended the melt peak into autumn. This effect was simulated to diminish with then decreased total streamflow in late summer. During autumn and winter, simulated streamflow for 2070–2099 was

nearly double the level of the historical period mainly due to an increase in the rainfall component of streamflow. Despite similar tendencies of reduced $Q_S$ in the future, differences arising from the different bias correction methods are notable. $Q_S$ was more prominent in all regimes based on MBCn-corrected GCM–RCM outputs, which simulated higher peaks during the snowmelt season and a generally higher fraction during the rest of the year, especially for the future periods. Accordingly, QDM-corrected data supported a larger $Q_R$ component beyond the summer. As a consequence, during low flow periods in winter, QDM-corrected forcing data overestimated the streamflow in the historical reference period. In contrast, QDM-corrected forced simulations tended to slightly underestimate the streamflow during the spring and summer months, as $Q_S$ was underestimated. Generally, MBCn-corrected data matched more closely with the reference simulations based on observed data.

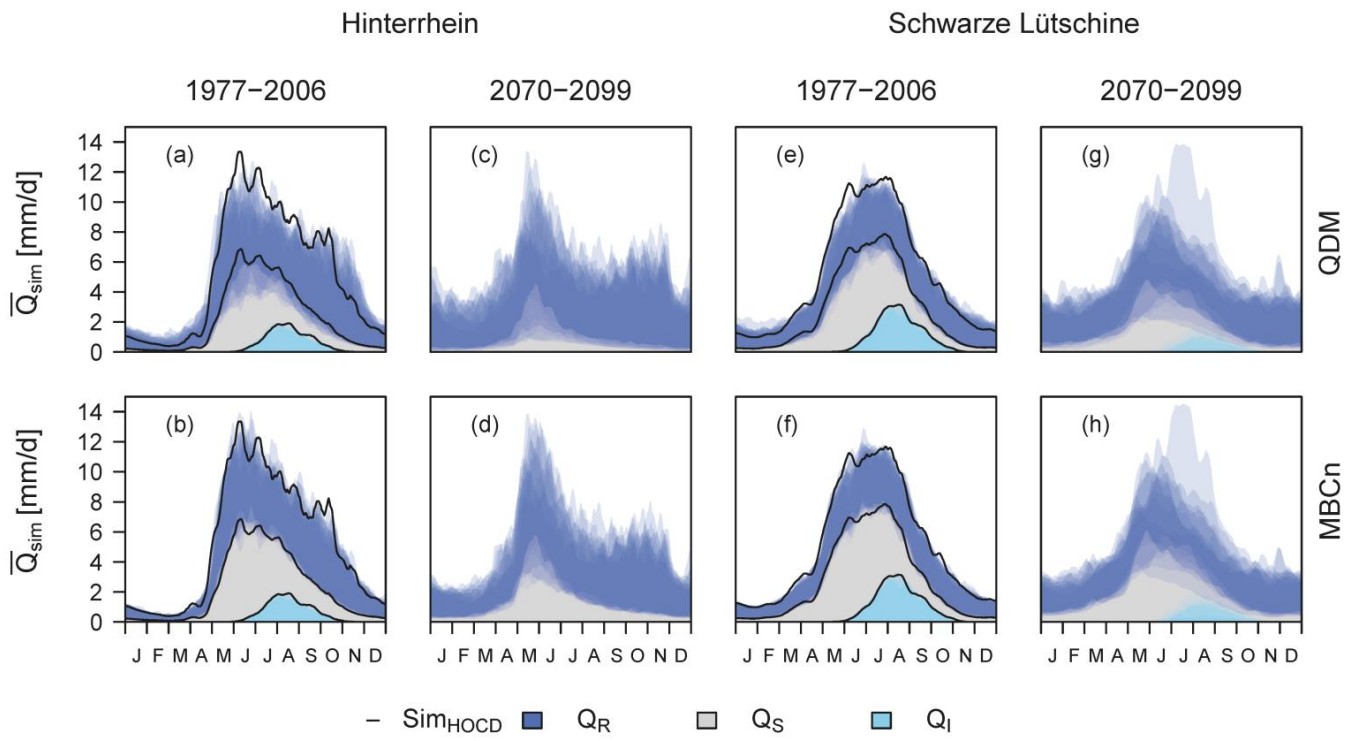

Figure 6: Streamflow regimes based on 11-day moving averages of daily streamflow during 30-year periods in the historical reference period and as projected for the period 2070–2099 under the RCP 8.5 scenario for the two catchments. Simulation results for each ensemble member are shown as semi-transparent polygons. For the historical reference period also the results of the simulations based on the historical reference $P$ and $T_a$ time series are shown (black lines).

## 5 Discussion

Both bias correction methods employed in this study, univariate QDM (Cannon et al., 2015) and multivariate MBCn (Cannon, 2018), are based on the same quantile mapping approach and by definition the marginal distributions of the corrected $P$ and $T_a$ series are the same as those of the historical reference data. However, the bias correction methods do result in differences in terms of $P$ and $T_a$ interdependency (see marginal and joint distributions of $P$ and $T_a$ series in the Supplement). Preserving the ranks of the climate model simulations, univariate bias correction approaches retain the inter-variable dependencies as represented in the raw climate model output (Vrac, 2018), as also demonstrated and discussed in previous studies using univariate quantile mapping methods (Wilcke et al., 2013; Ivanov and Kotlarski, 2017). However, often observed inter-variable dependencies are misrepresented in climate model simulations and hence biases therein are also retained by univariate methods (Wilcke et al., 2013; Gennaretti et al., 2015; Zscheischler et al., 2019). Such biases in the interdependency representation were also found in this study for $P$–$T_a$ interdependency of raw climate model output from ten GCM–RCM simulations compared to the used historical observational dataset (see also Supplement). In snow-dominated environments, the representation of precipitation–temperature interdependence is important for hydrological modelling but also for many other aspects impacted strongly by snow cover extent and duration (Gennaretti et al., 2015). Further studies that compare $P$–$T_a$ representation in climate model output and multiple observational datasets are needed to explore the causes of differences between climate model output and reference data such as those found here.

As air temperature determines the distinction between liquid precipitation and snow, differences in the climate variables' interdependence can lead to differences in simulated snowfall (Fig. 2), and consequently in snow accumulation and the catchments' seasonal water storage (Fig. 3–6). For the MBCn-corrected data in this study there was clearly more precipitation at air temperatures below 0 °C in comparison to the QDM-corrected data, resulting in more precipitation falling as snow, being stored, and accumulated than for univariate bias corrected forcing data. In glacierized catchments the higher amounts of snow from MBCn compared to QDM also affected the glaciers with higher winter mass balances and a later start of the melt season in spring/summer. The existence or non-existence of water storages in the form of snow and ice as well as the liquid precipitation directly contributing to streamflow had notable influences on the streamflow composition and regime. For instance, the larger fraction of liquid precipitation at the cost of snow simulated with QDM-corrected data led to a systematic overestimation of streamflow during the winter months in the historical reference period. This error was not present in simulations based on MBCn-corrected $P$ and $T_a$ forcing.

It bears noting that results from QDM and MBCn in the historical reference period are, as for example also in Zscheischler et al. (2019), evaluated without cross-validation. However, because the univariate and multivariate bias correction algorithms are applied in an asynchronous fashion to freely running climate simulations – adjusting the marginal/joint distributions – it is, by construction, almost guaranteed that they will perform well in terms of cross-validated measures of distributional fit. Cross-validation does make sense when performance – especially for aspects not explicitly adjusted – is measured in a setting where climate model simulations are synchronized with the real-world climate state, for example in climate

prediction or perfect boundary condition (e.g., reanalysis-driven) setups. We note that such reanalysis-driven cross-validation experiments have been performed in Cannon (2018) for the two algorithms used in this study. This was done over a large continental domain for a complicated multivariate fire weather index that combines, in a nonlinear fashion, the current and lagged effects of air temperature, precipitation, wind, and humidity. Hence, it is expected that results reported

here are robust and would be similar in an out-of-sample evaluation.

There have long been concerns over climate change impacts on mountain water towers. Many climate impact studies for snow-dominated catchments agree that due to continued warming, a decrease in snow cover characteristics and time-shifted snowmelt contributions to streamflow are to be expected under climate change scenarios (e.g. Barnett et al., 2005; Farinotti et al., 2012; Köplin et al., 2014; Addor et al., 2014; Milano et al., 2015; Coppola et al., 2016; Jenicek et al., 2018; Hanzer et

al., 2018). In fact, the shift and decrease of the snowmelt peak are one of the most robust results of such studies. In this study we showed that the snow component strongly depends not only on the GCM–RCM outputs but also on whether the bias correction method applied incorporates inter-variable dependence of $P$ and $T_a$ or not. The simulated glacier volume showed a clearly decreasing trend over the scenario period. However, net mass balances and hence rates of glacier ice melt and the mean timing of the final glacier disappearance vary by over a decade in the Hinterrhein catchment. While the ensemble

covers a wide range, the bias correction approach makes a difference for each GCM–RCM forcing. The changes in snow accumulation and glacier melt then propagate into changes of streamflow regimes. In future projections, snowmelt peaks tend to occur earlier and with a more concentrated melt season. A potential effect of this storage shift on streamflow however is potentially relevant year-round as could be visualized by the specific streamflow component modelling. The simulations suggest that the melt contribution to streamflow depends on the interdependence of air temperature and

precipitation and hence the chosen bias correction method. Furthermore, streamflow during the late summer decreases as the release of stored water from glaciers, which makes up a notable percentage of streamflow during the late summer, will have diminished. These systematic differences in hydrological impact scenarios originating from the applied univariate or multivariate bias correction method such as those found here, e.g. differences in glacier disappearance dates or differences in seasonal (summer vs. winter) water availability, may appear negligible given the overall large uncertainties of climate impact

modelling yet may still be relevant for some specific adaptation management questions. The timing of 'peak water' occurrence or complete disappearance of glaciers may be relevant for the planning horizon of hydropower schemes (Hänggi and Weingartner, 2012; Schaefli, et al., 2019). The earlier recession of the melt peak may sooner or later affect early-summer flood hazard or increase the hazard of late-summer low flow due to the loss of ice and snow components of streamflow (Beaulieu et al., 2012; Godsey et al., 2014) requiring the planning of respective measures.

These results also require discussion of implications on common conceptual hydrological modelling concepts that are needed to simplify meteorological and hydrological complexity. The use of a threshold air temperature for the distinction of precipitation in snow and rainfall is a key concept of the HBV model and many other hydrological models. Hence, it may be expected that the simulations of the snow-dominated catchments respond particularly sensitive to changes and biases in $P–T_a$ interdependencies. The question is the degree to which this may influence the hydrological variables discussed above. So far,

few studies have evaluated multivariate-corrected GCM–RCM data in hydrological modelling. Chen et al. (2018) found that the joint bias correction of precipitation and air temperature led to a much better performance in terms of hydrological modelling for all their study basins located in various climates except for the coldest Canadian basin. In contrast, an overall additional benefit of using bivariate bias correction methods for hydrological impact projections was not evident in results by

Räty et al. (2018) when compared to using a univariate quantile mapping applied as a delta change method, i.e. retaining present-day correlation structures. However, their analysis indicated that the selection of the bias correction method was most important and the added value of using multivariate approaches most clearly found for SWE simulations, supporting the findings of this study. Based on these case studies, it may be assumed that simulations with any hydrological model that include calibration over a historical reference period will be somewhat affected by a biased representation of inter-variable

dependence of its input variables in GCM–RCM outputs. Further studies are needed to investigate other effects of multivariate bias correction for other types of climatological input variables, hydrological models, catchment types, and dominating processes.

This study demonstrates the importance of considering the representation of the interdependence of precipitation and air temperature in the specific case of hydrological impact modelling of snow and glacier dominated catchments. As shown, in

the representation of the climate variables' interdependence, the multivariate bias correction approach leads to results closer to the climatological historical reference data as well as partly to hydrological simulations closer to the historical reference simulations as for instance for the simulated glacier volumes. Cannon (2016, 2018) also demonstrated better results for multivariate-corrected data in other examples, including fire weather indices and atmospheric river detection. In practice, some kind of bias correction is needed for many impact studies, although it is known that recent literature is rich in

controversial debate of its use and major limitations of the application of empirical-statistical bias correction methods (e.g. Ehret et al., 2012; Addor and Seibert, 2014; Maraun, 2013, 2016; Clark et al., 2016; Maraun et al., 2017; Casanueva et al., 2018; Zscheischler et al., 2019). Some of the fundamental issues, the details of which are beyond the scope of this study, are shared with univariate bias correction, for example, the question of stationarity (regarding biases in marginal distributions). In addition, joint correction is often based on the assumption that the structure of the bias in variables' interdependence is

stationary, i.e. the same for control as for projections. This is not strictly true for MBCn, which allows the multivariate distribution to evolve in the projection period. However, the extent to which model projected changes in dependence structure are preserved by MBCn have yet to be evaluated closely. More generally, whether the preservation of inter-variable dependence structures is a robust assumption or dependence structures should evolve from the reference to the future period are still open questions for multivariate bias correction methods development (Vrac, 2018). Furthermore, the correction of

the multivariate dependence structure will necessarily affect the time sequencing of the climate model variables (Cannon, 2016), which can lead to modification of temporal autocorrelation. Maraun (2016) cautions that modifications of spatial, temporal or multi-variable interdependence may break the consistency with the driving climate model and many others have argued for the least possible transformation of GCM–RCM outputs for this reason. This study does not address these fundamental questions and critiques nor does it generally recommend or not recommend the use of multivariate bias

correction methods. The objective of the study was to compare the differences resulting from univariate vs. multivariate methods. We demonstrated a case in which biases in inter-variable dependencies can affect hydrological simulations considerably. This is important, particularly as it is common practice to use hydrological models calibrated to climatic conditions represented by historical climate variable series. In the same way that the use of several climate and hydrological

models is recommended, the incorporation of uncorrected, univariate-, and multivariate-corrected scenario data in the ensemble may be considered as one part of a transparent and honest communication of the full range of uncertainties.

## 6 Conclusions

This study systematically tested the effects of multivariate bias correction of projected air temperature and precipitation versus a traditional univariate bias correction on hydrological impact modelling in alpine environments. Jointly corrected air

temperature and precipitation series simulated more snowfall and consequently up to 50% more snow accumulation than univariate-corrected GCM–RCM data. Subsequently, glacier volume was simulated to decrease by up to a decade slower under multivariate-corrected scenarios. These differences also impact the simulations of streamflow and its components with higher snowmelt components and accordingly smaller rainfall components under multivariate-corrected scenarios compared to univariate-corrected scenarios. These are relevant systematic differences despite variations of the GCM–RCM ensemble.

The choice between a univariate and a multivariate bias correction approach may therefore have implications for future water resources planning, as the snow component presents an important seasonal storage, and for the protection against hydrological hazards such as a higher vulnerability to drought.

Beyond the specific case this study suggests that the effect of bias correction methods may be generalized for catchments that include the elevation range of the snow line. Mountain hydrology modelling relies on the correct representation of the

interdependence of air temperature and precipitation due to a crucial role of threshold air temperature concepts for the distinction of liquid and solid precipitation. This study makes an argument for the explicit consideration of interdependencies of climate variables by using multivariate bias correction methods in hydrological climate change impact studies in snow-dominated catchments. But also many other threshold effects drive relevant climate impacts and are parameterized in many models or indices. The study provides a strong incentive to test similar effects in hydrological systems and their model

representations that may be dominated by other climate variable interdependencies.

## Code availability

An R package (R Core Team 2015) including the MBCn and the QDM algorithm is available for download from https://CRAN.R-project.org/package=MBC. The HBV-light software is freely available for download from https://www.geo.uzh.ch/en/units/h2k/Services/HBV-Model.html.

**Data availability**

EURO-CORDEX data can be accessed via different European datanodes, available at https://www.euro-cordex.net/060378/index.php.en. The HYRAS interpolation product used to derive the historical reference climate time series was made available by the German Weather Service (DWD) and the German Federal Institute of Hydrology (BfG). Streamflow time series were provided by the Swiss Federal Office for the Environment (FOEN) and the Amt für Wasser und Abfall des Kantons Bern. Snow data of the "SLF-Schneekartenserie Winter 1972-2012" used for model calibration are available upon request by the WSL Institute for Snow and Avalanche Research (SLF). Glacier ice thickness data were provided by Matthias Huss, other glacier data are available according to the given references.

**Author contribution**

JM, IK, KS, and JS designed the study. JM carried out bias correction, modelling, and all analyses and wrote the first draft. IK calibrated the hydrological model and prepared snow, glacier, and hydrological data. KH prepared the EURO-CORDEX data for the catchments. AC provided and helped with his bias correction scripts. All co-authors contributed to and edited the manuscript.

**Competing interests**

The authors declare that they have no conflict of interest.

**Acknowledgements**

Work for this study was based on data acquired and methods developed within the project 'The snow and glacier melt components of the streamflow of the River Rhine and its tributaries considering the influence of climate change' (ASG-Rhein, see Stahl et al., 2017) funded by the International Commission for the Hydrology of the Rhine basin (CHR). We thank Dr. Urs Beyerle for his assistance with the retrieval of EURO-CORDEX data and further thank all data providers (see 'Data availability'). The article processing charge was funded by the German Research Foundation (DFG) and the University of Freiburg in the funding programme Open Access Publishing.

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
