# Peer review of "Effects of univariate and multivariate bias correction on hydrological impact projections in alpine catchments"

_Hydrology and Earth System Sciences, 2018_

## Referee Comment (RC1) · O. Rössler (Referee) · 19 Jul 2018

The manuscript "Effects of univariate and multivariate bias correction on hydrological impact projections in alpine catchments" by Meyer et al. investigates the effect of univariate versus (BC) multivariate bias correction (MBC) on the representation of snow, ice and rain representation in a climate change impact assessment approach. MBC has the advantage to control for variable interdependency (T and P namely) that in turn influences rain-snowfall fractioning. They used BC and MBC to bias correct and downscale 10 GCM-RCM combination using quantile mapping and drove the hydrological model HBV-light in a transient setting (1976-2099). They analyzed the effect of

the different methods for snow-water-equivalent, icemelt, streamflow amounts and its composition (ice, snow, rain) over time. The paper adds to the ongoing discussion on the possible effects of intervariable dependency, especially by adding the information about the effect of streamflow composition.

The paper is scientifically interesting, original, overall very well writing and certainly within the scope of the journal and of interest to the readers. Texts and figures are widely clear and lead the way for reasonable conclusions.

Beside some minor comments that I will list below, I have two more substantial concerns that refer to the design of the study.

1. The hydrological model was calibrated for the entire reference time period, e.g. 1976 – 2006, (page 7, line 16 ff) against streamflow, snow (SWE and snow-covered area), and glacier volume.

Three things puzzles me here:

1.1 There is no validation period!

1.2 Model performances against observation are not presented at all, neither as statistical measure nor in the graphs. I am aware that only differences between input data sets are analyzed in this study, still the basic performance measures are needed to frame the results. E.g. if the representation snow melt is not well captured (what I do not assume here) but the streamflow is (hence snow melt insensitive) than the snow sensitivity to changes in the input data might also be underrated.

1.3 this concern about validation and performance measures also extends to both QM approaches (was a cross-validation framework used? provision of verification statistics is needed)

–> more emphasis should be laid on the presentation of validation and the introduction of a validation period/validation framework is required

2. If I understand correctly, the combination of climate model data and the hydrological model is as follows:

- The quantile mapping is performed between climate model output and the average catchment value for T and P

- This mean value is interpolated within the catchment by a lapse rate that is fixed for each day of the year (extracted from the reference period)

This approach might be needed in HBV-light, but is based on the assumption that the lapse rate is not changing over time and is independent of certain events. This is a very strong assumption that is disproven by numerous study showing e.g. elevation depended warming, for instance. I assume, and to my own experience, that the slope of the lapse rate is quite sensitive to SWE simulations. Hence, this strong assumption likely influences the robustness of your results. Furthermore, you cannot control for this in the calibration of the model, as you limited the evaluation of model performances of SWE and snow-covered area to 2000 – 2500 m asl (page 7, line 21), which is exactly the catchment mean elevation for which the lapse rate is of minor effect. To me the fixed lapse rate is a very critical points in the study and need to be solved.

A possible workaround of this issue would be to perform a quantile mapping that is not based on catchment mean values, but for each grid cell of the HYRAS data set. This procedure is currently done for the new CH2018 climate scenarios by MeteoSwiss. Doing so, you can extract the lapse rate for each day separately and use this dynamic lapse rate. With this procedure you would not have to make this strong assumption of a static lapse rate and have much more reliable results.

I think that these proposed changes are accomplishable in a reasonable time. Therefore, if those and the following comments are addressed, I am happy to comment on the manuscript again and likely recommend a publication. I am looking forward to the revised version.

Specific comments:

Page 1, line 23ff. I suggest to state this result ("for the historical..:") prior to the effect on the future as this ensures an improved bias correction for MBC.

Page 2, line 24: This publication might also be of interest (I am not an author): Wilcke, Renate Anna Irma; Mendlik, Thomas; Gobiet, Andreas (2013): Multi-variable error correction of regional climate models. In: Climatic Change 120 (4), S. 871–887. DOI: 10.1007/s10584-013-0845-x.

Page 4 line 2: Only the "Unterer Grindelwald"-glacier is big (∼6.biggest in Switzerland). It is a glaciated catchment, but covered by smaller glaciers. Please, rephrase. Therefor, also the following sentence needs to be rephrased.

Page 5, line 7ff: Please highlight that you apply catchment averages and that these averages are the "Target" in the quantile mapping approach (If I understood you correctly)

Page 5, line 10: This is a very unusual time period as it crosses to climate normal periods. Do you have any reason of this time window. It hampers comparability to other climate change impact assessment studies.

Page 5, line 13. Which gauging station was used. I am only aware of the FOEN station in Lütschine-Gsteig, and the Weisse Lütschine, Zweilütschinen. Did you use differences of these stations?

Page 6, line 6: This is phrased wrongly. Univariate QDM cannot be both widely accepted (and used since several years) and developed by Cannon et al. 2015.

Page 6, line 9: detrending of a time series is problematic, as assumptions about the kind of trend are necessary. Can you please add information about the way the trend is treated and comment on possible effects.

Page 6, chapter 3.1: additional information about the validation procedure should be

given as well as information about the "target value" (catchment averages)

Page 6, line 22: "Climate model data" instead of data

Page 6, line 28: Please, quantify the difference by adding grid cell size and range of catchment area.

Page 7, line 16: so no separation of calibration and validation time period ! see general comments

Page 7, line 21-22: is it correct that you not only evaluate but calibrate your model against this elevation limitation? Please rephrase.

Page 7, line 21-22: I disagree with the statement that only the area 2000-2500 is crucial as in my experience it is also very important for streamflow how much of the entire catchment is covered by snow – and hence contribute to snow melt. Please, comment.

Page 8, Figure 2: What is also striking is that noBC is performing better than QDM for rainfall. Can you add on this?

Page 10, Figure 4: Maybe an error occurred, as the boxplots in the lower panel of the Schwarze Lütschine graph is missing.

Page 14, line 22: Please rephrase: It depends not on the bias correction but more specifically on the consideration of intervariable dependencies.

Page 14, line 24: Are the found glacier retreat comparable to other findings?

Page 15, line 8: Much more critical to me is the assumption of a fixed lapse rate, even more under climate change conditions

Page 15, line 34: is there a type? "re bias "

---

## Author Comment (AC1) · 21 Aug 2018

**Authors' response to the interactive comment of referee # 1 on hess-2018-317**
"Effects of univariate and multivariate bias correction on hydrological impact projections in alpine catchments" by Judith Meyer et al.

We thank the reviewer Ole Rössler for his in general positive evaluation of our manuscript, and the many helpful comments. Below we respond (in blue) to the reviewer comments (*in black*). We appreciate the efforts by the reviewer, which will help to improve the clarity of our manuscript.

*The manuscript "Effects of univariate and multivariate bias correction on hydrological impact projections in alpine catchments" by Meyer et al. investigates the effect of univariate versus (BC) multivariate bias correction (MBC) on the representation of snow, ice and rain representation in a climate change impact assessment approach. MBC has the advantage to control for variable interdependency (T and P namely) that in turn influences rain-snowfall fractioning. They used BC and MBC to bias correct and downscale 10 GCM-RCM combination using quantile mapping and drove the hydrological model HBV-light in a transient setting (1976-2099). They analyzed the effect of the different methods for snow-water-equivalent, icemelt, streamflow amounts and its composition (ice, snow, rain) over time. The paper adds to the ongoing discussion on the possible effects of intervariable dependency, especially by adding the information about the effect of streamflow composition. The paper is scientifically interesting, original, overall very well writing and certainly within the scope of the journal and of interest to the readers. Texts and figures are widely clear and lead the way for reasonable conclusions.*

*Beside some minor comments that I will list below, I have two more substantial concerns that refer to the design of the study.*

In summary, as we understand, these more substantial concerns are: a) hydrological modelling results for a validation period are missing, and b) the use of a constant temperature lapse rate in HBV. We provide detailed explanations and suggestions for compromises below. But summarised in brief, these decisions need to be seen in context with the aim of the study to test principal effects of uni- vs. multivariate bias correction in such an environment. The objective of the study is not to present a best possible hydrological impact assessment for the two case study catchments (and result should not be taken as that, i.e. interpreted quantitatively as future scenarios). The case study catchments (rather than e.g. a completely hypothetical catchment, which would be an alternative) allow demonstrating and discussing the potential effects; i.e. the objective is to compare simulation results for the same boundary conditions in terms of hydrological modelling only differing in the $T_a$ and P input series corrected with a uni- vs. a multivariate method. In the revised version we will state this more clearly in the aims.

The use of the set lapse rate is partly due to the small scale of the catchments and HBV's tradition. We argue that both do not affect our conclusions regarding the specific aims of this study, in which focus is on comparison of systematic effects of bias correction methods, to keep the paper concise, and modelling with a conceptual model that is as parsimonious as possible. Note that Figure 2 (P at $T >$ or $\leq 0$ °C) compares results of QDM vs. MBCn independent of the hydrological modelling and related assumptions. Subsequent results of hydrological modelling are all reasonable consequences of that. Changes in the hydrological model parametrisation or incorporation of changed lapse rates may have led to slightly changed simulation results but we are convinced that that would not lead to different main conclusions in the comparison of the simulations driven by univariate and multivariate corrected P- and T-input.

*1. The hydrological model was calibrated for the entire reference time period, e.g. 1976–2006, (page 7, line 16 ff) against streamflow, snow (SWE and snow-covered area), and glacier volume.*

*Three things puzzles me here:*

*1.1 There is no validation period!*

*1.2 Model performances against observation are not presented at all, neither as statistical measure nor in the graphs. I am aware that only differences between input data sets are analyzed in this study, still the basic performance measures are needed to frame the results. E.g. if the representation snow melt is not well captured (what I do not assume here) but the streamflow is (hence snow melt insensitive) than the snow sensitivity to changes in the input data might also be underrated.*

The reason for not reserving a validation period was to make full use of available observation-based data, in particular on glacier change. As indicated by the referee, we think that details of model performance and hydrological model validation are not of highest priority to compare the results of univariate and bivariate bias correction of precipitation and temperature in hydrological modelling of alpine catchments in a general way, as we intend with our study. However, we fully agree that model validation is an important component of hydrological modelling studies in general and that some information on model performance should be added as general information for the reader.

1.1. Validation period: In our case the historical reference period 1976–2006 used for model calibration was chosen because (i) the used HYRAS climate data product unfortunately covers only the period ≤ 2006, and (ii) model warmup started in the year 1973, the simulation in 1976 and needs to go until at least 2003 in order to capture the climatology and to make full use of reliable data for the initial conditions and development of the catchments' glaciers. We consider the use of observation-based glacier data for both, model initialisation and calibration, crucial. Hence, we used an estimate of glacier volume for the year 2003 based on glacier area (by Paul et al. 2010) in model calibration. Consequently, a calibration period of at least 1973/1976 to 2003 has been a result of the available glacier data for the years of 1973 and 2003. Our model calibration strategy was developed within a multi-catchment modelling study (Stahl et al. 2017), for which a consistent calibration scheme and consistent reference datasets for 49 alpine catchments (including their glaciers) were needed. We are not aware of any additional glacier observation-based data covering all glacierised parts of the modelled catchments Hinterrhein and Schwarze Lütschine that could have been used additionally in this study.

We agree that validation normally should be done but was not used here because the study rather focused on the relative performances of the bias correction approaches than on the absolute model performance. We will stress this more clearly in the revised manuscript. However, to address the reviewer's request for a validation period, we suggest that we will re-calibrate the model until the year 2003 only, which would at least allow three years (2004–2006) to be used for model validation.

1.2 Model performance: Performance is usually far from perfect. But considering that we deal with high-elevation catchments, incorporate streamflow, snow characteristics, and glacier data in calibration we consider it an acceptable real-world representation for the purpose of the study. To address the reviewer's comment, we suggest to add quantitative results on model performance for the calibration period based on the components of the multi-criteria objective function and for the new validation period 2004–2006 to the revised manuscript. If required, we suggest some obs. vs. sim.

graphs as supplementary material to not complicate existing results graphs in which the focus should be kept on the comparison of QDM- and MBCn- based results.

*1.3 this concern about validation and performance measures also extends to both QM approaches (was a cross-validation framework used? provision of verification statistics is needed) –> more emphasis should be laid on the presentation of validation and the introduction of a validation period/validation framework is required*

The BC methods are optimized to the best possible transformation. We think, a randomized cross-validation would not provide any targeted and usable information regarding our aims of testing systematic effects of QDM vs MBCn. Because the univariate and multivariate bias correction algorithms are applied in an asynchronous fashion to freely running climate simulations – adjusting the marginal/joint distributions – it is almost guaranteed that they will perform well in terms of cross-validated measures of distributional fit. This is by construction. See also the critique by Maraun & Widmann in HESSD: https://www.hydrol-earth-syst-sci-discuss.net/hess-2018-151/.

Cross-validation does make sense when performance, especially for aspects not explicitly adjusted, is measured in a setting where climate model simulations are synchronized with the real-world climate state, for example in climate prediction or perfect boundary condition (e.g., reanalysis-driven) setups. We note that such reanalysis-driven evaluations have been performed in Cannon (2018) for the two algorithms used in this study. This was done over a large continental domain for a more complicated multivariate index (fire weather), that combines, in a nonlinear fashion, the current and lagged effects of temperature, precipitation, wind, and humidity. To address the reviewer's request, we suggest that these issues and past verification efforts be discussed briefly in the revision.

*2. If I understand correctly, the combination of climate model data and the hydrological model is as follows:*
*- The quantile mapping is performed between climate model output and the average catchment value for T and P*
*- This mean value is interpolated within the catchment by a lapse rate that is fixed for each day of the year (extracted from the reference period)*

*This approach might be needed in HBV-light, but is based on the assumption that the lapse rate is not changing over time and is independent of certain events. This is a very strong assumption that is disproven by numerous study showing e.g. elevation depended warming, for instance. I assume, and to my own experience, that the slope of the lapse rate is quite sensitive to SWE simulations. Hence, this strong assumption likely influences the robustness of your results. Furthermore, you cannot control for this in the calibration of the model, as you limited the evaluation of model performances of SWE and snow-covered area to 2000 – 2500 m asl (page 7, line 21), which is exactly the catchment mean elevation for which the lapse rate is of minor effect. To me the fixed lapse rate is a very critical points in the study and need to be solved. A possible workaround of this issue would be to perform a quantile mapping that is not based on catchment mean values, but for each grid cell of the HYRAS data set. This procedure is currently done for the new CH2018 climate scenarios by MeteoSwiss. Doing so, you can extract the lapse rate for each day separately and use this dynamic lapse rate. With this procedure you would not have to make this strong assumption of a static lapse rate and have much more reliable results.*

Firstly, for clarification: calibration in terms of snow covered area fraction was based on the entire catchment only the calibration of SWE was limited to the elevation range 2000–25000 m asl. Phrasing of the sentences referring to the calibration of snow characteristics is not wrong but a bit misleading and will be improved in the revised version. If we re-calibrate the model for the revisions to have a validation period (comment above) we'll consider to use a larger elevation range for the SWE calibration (see also separate comment on SWE calibration below).

We agree that lapse rates and their potential future changes are important for hydrological modelling of alpine catchments in particular and that hydrological impact assessments should attempt to incorporate them adequately. However, this is not easy and a topic related to high uncertainties anyway. The use of a linear gradient is a key feature of HBV. That is already a – if you like, strong, – simplification as one of many common in hydrological modelling.

We considered the possible ways to derive bias corrected lapse rates from GCM–RCM output:

- Referee's suggestion to apply BC on a grid-to-grid base to HYRAS: this would in fact be an application of BC to a grid distinctly higher in resolution (1x1 km) that is, as we mentioned in the paper, also criticised widely and would call for an explicit downscaling step, i.e. it would require aggregation and re-gridding of HYRAS. Consequently, results would also be strongly influenced by the underlying background climatology of this HYRAS interpolation product (also available only for recent conditions). We are not convinced that this is an appropriate/better solution.

- Theoretically, of course it is feasible to extract lapse rates from the GCM–RCM output and incorporate in the bias correction, including in case of the multivariate MBCn to apply it to 4 variables, not only to T and P catchment mean but additionally to lapse rates. Uncertainties in the extraction of lapse rates from 12x12 km CORDEX grid for mesoscale catchments (54 km² the smaller one) and their subsequent bias correction are expected to be substantial (even higher than those in case of the catchment averages), too. In addition, we think that evidence of trends in projected temperature lapse rates is limited at the catchment scale.

➜ Considering our aims and setting (model, data, objectives) we don't think that extra efforts vs. expected added value at the expense of more assumptions may improve the confidence in the conclusion of the study. We think it would rather complicate the interpretation of the results in terms of effects of the QDM vs. MBCn on $T_a$ and P. We prefer simple, comparable boundary conditions for our modelling experiment, and advocate keeping lapse rates unchanged/unaffected. As stated above, we'll stress more clearly that our study should be taken as that, a modelling experiment not a prefect impact assessment and that taking the long-term seasonal average lapse rate pattern from observed data as stationary was one of our assumptions to facilitate the separation of the effect of incorporating or ignoring interdependencies of catchment averages of $T_a$ and P in bias correction in the interpretation of results.

*I think that these proposed changes are accomplishable in a reasonable time. Therefore, if those and the following comments are addressed, I am happy to comment on the manuscript again and likely recommend a publication. I am looking forward to the revised version.*

To summarise briefly: we are going to re-calibrate for 1976–2003 with validation for 2004–2006, add information on performance, and we will reconsider the SWE calibration (probably extent to a larger elevation range). While not crucial for our study's main objective, those revisions are straightforward

and time efforts justifiable. But from our side and with the study's objective in mind we do not see the need for cross-validation of bias correction and argue to keep fixed lapse rates (see above).

*Specific comments:*

*Page 1, line 23ff. I suggest to state this result ("for the historical..:") prior to the effect on the future as this ensures an improved bias correction for MBC.*

The order of the sentences in the abstract referring to results will be changed in the revised version (first historical then future). Thanks.

*Page 2, line 24: This publication might also be of interest (I am not an author): Wilcke,Renate Anna Irma; Mendlik, Thomas; Gobiet, Andreas (2013): Multi-variable error correction of regional climate models. In: Climatic Change 120 (4), S. 871–887. DOI:10.1007/s10584-013-0845-x.*

Thanks for this suggestion. We will check this reference and add here if appropriate. However, note that we are aware that several studies, as this one, argue that applying (univariate) quantile mapping retains the inter-variable features of the driving model. This is not surprising because quantile mapping is a monotonic transformation; by construction, it cannot change the temporal rank ordering of the climate model variables and, as a result, will retain the climate model's inter-variable rank correlation structure. Multivariate methods, by extension, can modify the inter-variable dependence structure so that it more closely matches that of the target observations. We may briefly address this in the discussion of the revised manuscript.

*Page 4 line 2: Only the "Unterer Grindelwald"-glacier is big ( ~ 6.biggest in Switzerland). It is a glaciated catchment, but covered by smaller glaciers. Please, rephrase. Therefor,also the following sentence needs to be rephrased.*

We agree, thanks. The sentences will be rephrased in the revised version

*Page 5, line 7ff: Please highlight that you apply catchment averages and that these averages are the "Target" in the quantile mapping approach (If I understood you correctly)*

Yes, we work generally with series of catchment averages also as "target" data as stated already (clearly?) at the beginning of the data section Page 4 line 16: "The resulting time series of catchment mean precipitation and temperature were used as input for the calibration of the glacio-hydrological model and as historically observed climate data (HOCD) for the bias correction." We'll try to stress this even more clearly in the revised manuscript.

*Page 5, line 10: This is a very unusual time period as it crosses to climate normal periods. Do you have any reason of this time window. It hampers comparability to other climate change impact assessment studies.*

Use of the period 1976–2006 was motivated by the modelling of glacier change. Glacier data for the catchments (all glacierised parts) were available for the years 1973 (glacier area based on the inventory by Müller et al. 1976 / Maisch et al. (2000) and modelled ice thickness data provided by

Matthias Huss) and 2003 (inventory by Paul et al. 2010). In our opinion, the initialisation of the glaciers represents a major source of uncertainty for modelling of glacierized catchments. Hence, we wanted to start the simulations (warmup) at a year for which glacier- area and -volume distributions are relatively well constrained by observation based data. Furthermore, we think it is important to incorporate glacier data in model calibration, what we realised by using glacier volume estimates based on glacier area data for the year 2003 (see also comment above). The used HYRAS data product is currently updated but was not available for periods beyond 2006 at the time of the study and is not yet. Moreover, we would like to stress again that our study should be mainly considered as a general test to reveal and discuss principal "effects of univariate and multivariate bias correction" using two alpine catchments for demonstration rather than a specific climate impact assessment for the two case study catchments. We think, we clearly focused our interpretation of results on the comparison of univariate and bivariate bias correction; it is only limited possible to discuss the presented simulation results quantitatively in terms of projected impacts and to compare them with more comprehensive climate impact assessments.

*Page 5, line 13. Which gauging station was used. I am only aware of the FOEN station in Lütschine-Gsteig, and the Weisse Lütschine, Zweilütschinen. Did you use differences of these stations?*

Yes, exactly we used the records from the stations Lütschine–Gsteig and Weisse Lütschinen–Zweilütschinen to reconstruct streamflow 1973–2006 for the outlet of the study catchment (in mm/day) at the location of the station Schwarze Lütschine – Gündlischwand operated by the Canton of Bern. Observed stream flow at the station Schwarze Lütschine – Gündlischwand that was available for the period 1992–1999 was used to validate the reconstructed streamflow time series. We agree that this should have been explained in the manuscript and will add this in the revised version.

*Page 6, line 6: This is phrased wrongly. Univariate QDM cannot be both widely accepted (and used since several years) and developed by Cannon et al. 2015.*

Sentence will be rephrased to be more precise, by referring to univariate quantile mapping techniques in general and the specific QDM approach afterwards.

*Page 6, line 9: detrending of a time series is problematic, as assumptions about the kind of trend are necessary. Can you please add information about the way the trend is treated and comment on possible effects.*

The approach used for QDM follows the quantile delta change (Olsson et al. 2009, Atmos. Res.) and quantile perturbation (Willems and Vrac 2011, J. Hydr.) methods and is explained and discussed in detail in Cannon et al. (2015). Strictly speaking, the use of "detrending" here is inaccurate and needs to be reworded. The detrended quantile mapping (DQM) algorithm in Cannon et al. (2015) *does* use a linear detrending step, but this is not the case with the QDM (and, by extension, MBCn methods). QDM adjusts all quantiles in the projection period by first removing the projected change signal (e.g., if the climate model projects a 20% increase in the $90^{th}$ percentile of precipitation in the future period, this change is removed, etc.), then it applies standard quantile mapping, and finally the projected change signal is reintroduced. Aside from rewording, we may provide some brief explanations and hints on references in the revised version. However, we do not think that detailed explanations and discussion are needed within the scope of this study, given that this is included in the original paper on the method

*Page 6, chapter 3.1: additional information about the validation procedure should be given as well as information about the "target value" (catchment averages).* Please, see general comments above.

*Page 6, line 22: "Climate model data" instead of data* → Will be changed, thanks.

*Page 6, line 28: Please, quantify the difference by adding grid cell size and range of catchment area.*

Information on climate model data resolution (0.11°: ~12x12km) and catchment sizes (54 km² and 180 km²) are given in the corresponding subsections but may be repeated here (in brackets) in the revised version.

*Page 7, line 16: so no separation of calibration and validation time period ! see general comments*

Please, see response to general comment above, too.

*Page 7, line 21-22: is it correct that you not only evaluate but calibrate your model against this elevation limitation? Please rephrase.* See below for a combined reply to this and the next comment.

*Page 7, line 21-22: I disagree with the statement that only the area 2000-2500 is crucial as in my experience it is also very important for streamflow how much of the entire catchment is covered by snow – and hence contribute to snow melt. Please,comment.*

See also response to general comment above. For model calibration we used: a) snow covered area fraction of the entire catchment!, b) mean SWE for the elevation range 2000–25000 m a.s.l. Both were derived from a gridded SWE climatology product from the Swiss SLF institute. Due to discrepancies in the data product's resolution and the high detail of elevation-and-aspect zones for glacierised and non-glacierised catchment part for the catchment model, we think a comparison of mean values calculated for the entire catchment or a (not to limited) elevation range most appropriate. The reference data product from the SLF is known to be based on relatively few stations for elevations above 2500 m (in the domain of the study); thus, we constrained the use of the data to elevations below 2500. Selection of the elevation range of 2000–25000 was based on an analysis of the SLF product for SWE statistics over all elevation zones of 49 alpine catchments (alpine catchments modelled in Stahl et al. 2017). Focussing on all those catchments, we considered the elevation range 2000–25000 m a.s.l. particularly important. However, we agree that an extension of elevations in the catchments SWE calibration might be useful here. We will consider this for the planned re-calibration of the model for the revisions, likely extend the elevation range used for calibration of SWE, and will also explain the calibration for snow characteristics more clearly.

*Page 8, Figure 2: What is also striking is that noBC is performing better than QDM for rainfall. Can you add on this?*

This is not the case. If understood correctly, your statement is "true" if one looks at the median of the boxplots for rain for days with $T_a > 0$ °C only (left part of left panel plots for historical period) ignoring the considerable bias for rain for days with $T_a \leq 0$ °C (right part of left panel plots for historical period). Looking at total precipitation independent of $T_a$, considerable biases towards too large precipitation sums are evident in the noBC data (from all GCM–RCM combinations). Both

methods (univariate QDM and MBCn) correct for this to a similar degree, with marginal distributions the same for both methods, by definition, as also stated in the manuscript.

*Page 10, Figure 4: Maybe an error occurred, as the boxplots in the lower panel of the Schwarze Lütschine graph is missing.*

No, that is not an error but on purpose. As stated in the main text, for the Schwarze Lütschine catchments bias-corrected data from only few GCM–RCMs resulted in complete glacier retreat (volume = 0) by 2099. For those cases circles for the simulated year of complete glacier disappearance are plotted, whereas for most cases simulations resulted in a glacier volume ≥ 0 in 2099, thus no boxplot can be plotted. We will remark on that in the figure caption of the revised version.

*Page 14, line 22: Please rephrase: It depends not on the bias correction but more specifically on the consideration of intervariable dependencies.* Ok, sentence will be rephrased.

*Page 14, line 24: Are the found glacier retreat comparable to other findings?*

As stated above, we think one should discuss our simulations results with a focus on differences in the simulations (effects) as a result of the univariate vs. bivariate bias correction of temperature and precipitation rather than taking our results as specific future projections for the two catchments. We are not aware of projections specifically for the glaciers in our catchments that could be directly compared. Roughly, projections for mid-sized glaciers in the Aare basin might be compared with our simulations for the glaciers in the Schwarze Lütschine catchments and projections for small glaciers in the Alpine Rhine basin may be taken for comparison with our results for the glaciers in the Hinterrhein catchments. Results and discussion of projected glacier retreat and disappearance dates for small Swiss glaciers can be found in Huss & Fischer (2016, frontiers in earth science, https://doi.org/10.3389/feart.2016.00034); general results of projections for Swiss glaciers are presented, for instance, in the CCHydro report (FOEN, 2012: Effects of Climate Change on Water Resources and Waters. Synthesis report on "Climate Change and Hydrology in Switzerland" (CCHydro) project. Federal Office for the Environment, Bern. Umwelt-Wissen No 1217). In general, as far as a comparison is possible, the projected glacier retreat in our study compares well with results for projected glacier area change presented in those publications. However, note that i) it is generally difficult to compare the response and evolution of individual glaciers, ii) in our study we consider the difference between the simulations driven by QDM- and MBCn-corrected data of much higher relevance than the exact date of glacier disappearance in comparison to other studies.

*Page 15, line 8: Much more critical to me is the assumption of a fixed lapse rate, even more under climate change conditions*

Please, see general comments above. We agree that the question how to derive lapse rates for climate impact projections adequately is a difficult and an important challenge requiring research efforts. However, we restricted our study to demonstrating the effect of incorporating or ignoring interdependencies of catchment averages of $T_a$ and P in bias correction.

*Page 15, line 34: is there a type? "re bias "* Will be rephrased in the revised version.

---

## Referee Comment (RC2) · Kotlarski (Referee) · 3 Jan 2019

The work by Meyer et al. presents an inter-comparison of a univariate and a multivariate bias correction (BC) method in terms of hydrological climate impact scenarios in two catchments of the Swiss Alps. For this purpose, daily temperature and precipitation amounts as simulated by ten EURO-CORDEX RCM experiments are bias-corrected toward observed catchment mean values and then fed into the HBV-light hydrological model. For BC the QDM and MBCn methods are employed, the latter taking explicitly into account variable interdependencies. The study finds important differences in the simulated streamflow for a historical period between QDM- and MBCn-based setups.

In general, shows MBCn shows a better performance. The main reason is an underestimation of snowfall amounts in the QDM-based setups (equivalent to a smaller snowfall fraction of total precipitation) which translates into smaller SWE amounts and an overestimation of winter streamflow while the spring meltwater peak is underestimated. The differences in the snowfall amounts between the two BC approaches furthermore translate into differences in future climate change signals of SWE, glacier coverage and, finally, streamflow. Qualitatively, the differences between the BC approaches are obtained for all ten climate model chains investigated, indicating a robust finding that seems to be valid for any GCM-RCM chain.

In general, the paper is of very high quality and nicely highlights an important potential deficiency of bias-corrected climate scenarios in the Alpine region. It comes at a perfect time, as several recently released national reference scenarios are based on univariate BC approaches similar to QDM (e.g., Austria: ÖKS15, Switzerland: CH2018). As such, the study is certainly relevant for the journal's readership. Its setup is sound and convincing, the results are presented in an appropriate manner and the conclusions are well-based on the results obtained. There are no language issues except for the mixed use of past and present tense in the presentation of the results, which should be revised. There are a few minor issues that should be corrected for as well as two major remarks (see below). However, I'd leave it up to the authors to consider these major comments or not. I believe a consideration would further improve the quality of the paper, but the study is sound and convincing even in its current state. My recommendation is therefore to return the paper to the authors for minor revisions.

Congratulations for this nice piece of work! Sven Kotlarski

MAJOR ISSUES

Cross validation: Similarly to the point raised by a previous reviewer, I believe that a proper cross validation framework would be helpful. MBCn is a more complex method than QDM, and there's an increased danger of overfitting. As MBCn explicitly cor-

rects for biased inter-variable dependencies, snowfall amounts (if derived by a fixed temperature threshold) are well represented by definition. Being aware of the criticism by Maraun & Widmann, cross validation still makes sense in a split sample framework, e.g. by separating the historical period into the 15 coldest/warmest/driest/wettest years and the 15 warmest/coldest/wettest/driest years and using these subsets for calibration and verification, respectively. In case this splitting cannot be handled by HBV-light because the transient character is lost, one could carry out a cross validation for at least one ERA-Interim EURO-CORDEX experiment (these experiments are available as well and are in basic temporal correspondence with the observations). In general a cross validation would make the point stronger that a multivariate BC is superior for the example presented.

Reason for underestimated snowfall amounts by QDM: I understand that the paper puts an emphasis on the hydrological consequences of the two different BC methods. These effects are very well and convincingly presented. However, the question WHY QDM shows these deficiencies is not ultimately answered. The reason is to be found in the T-P relationship of the QDM data, and probably already appears in the raw RCM data. To analyze this further, 2D histograms would be extremely helpful and also illustrative.

MINOR ISSUES

Introduction and conclusions: The literature review should account for the studies by Wilcke et al. (Climatic Change, 2013) and Ivanov&Kotlarski (Int. J. Climatol., 2017). Inter-variable dependencies in standard QM have already been analyzed in there. One of the results was that QM does not distort inter-variable dependencies as long as they are approximately represented by the raw RCM data. The results of the present work therefore indicate some distorted inter-dependencies already in the RCM raw output (which could be better described if my major comment #2 would be considered). These issues should also be discussed in the discussion/conclusions.

p2 l20: "... which correct for biases in the data's entire distribution..."

[Figure]

p2 l4: The CORDEX data are actually not available from the CH2018 archive. The respective website only explains the selection of EURO-CORDEX models for the CH2018 Swiss climate scenarios. In the present study, EURO-CORDEX data were probably obtained from the ESGF archive.

Table 2: Just a note: The two runs driven by CNRM-CM5 are critical as the driving GCM CNRM-CM5 has an inconsistency in the historical period. It is fine to use them for the present work, but in future works they might have to be removed. More information is available from the new EURO-CORDEX errata page available from www.euro-cordex.net.

Chapter 3.1: The description of the QM methods is incomplete in the sense that it is not clear if the correction has been carried out for the bulk series (all days independent of the time of year) or depending on the time of year (e.g., seasonal or DOY dependence). This information is critical, as a bulk correction could be responsible for the deficiencies of QDM in my opinion. I believe the authors employed a seasonally dependent BC, but this needs to be better explained (even if reference to Cannon et al. is provided).

p6 l20-21: "... until the multivariate distributions of bias-corrected and observed data match."

p11 l30-31. Do you have any explanation for the higher mean streamflow amounts for QDM? Are differences in ETP involved?

p14 l24: "...disappearance vary by over a decade ..."

p15 l31: "...empirical-statistical bias correction methods ..."

---

## Author Comment (AC2) · 10 Jan 2019

**Authors' response to the interactive comment of referee # 2 on hess-2018-317**
"Effects of univariate and multivariate bias correction on hydrological impact projections in alpine catchments" by Judith Meyer et al.

We thank the reviewer Sven Kotlarski for his positive evaluation of our manuscript and the helpful comments. Below we respond (in blue) to the reviewer comments (*in black*). We appreciate the efforts of the reviewer and are going to consider his valuable suggestions when revising our manuscript.

*The work by Meyer et al. presents an inter-comparison of a univariate and a multivariate bias correction (BC) method in terms of hydrological climate impact scenarios in two catchments of the Swiss Alps. For this purpose, daily temperature and precipitation amounts as simulated by ten EURO-CORDEX RCM experiments are bias-corrected toward observed catchment mean values and then fed into the HBV-light hydrological model. For BC the QDM and MBCn methods are employed, the latter taking explicitly into account variable interdependencies. The study finds important differences in the simulated streamflow for a historical period between QDM- and MBCn-based setups.*

*In general, shows MBCn shows a better performance. The main reason is an underestimation of snowfall amounts in the QDM-based setups (equivalent to a smaller snowfall fraction of total precipitation) which translates into smaller SWE amounts and an overestimation of winter streamflow while the spring meltwater peak is underestimated. The differences in the snowfall amounts between the two BC approaches furthermore translate into differences in future climate change signals of SWE, glacier coverage and, finally, streamflow. Qualitatively, the differences between the BC approaches are obtained for all ten climate model chains investigated, indicating a robust finding that seems to be valid for any GCM-RCM chain. In general, the paper is of very high quality and nicely highlights an important potential deficiency of bias-corrected climate scenarios in the Alpine region. It comes at a perfect time, as several recently released national reference scenarios are based on univariate BC approaches similar to QDM (e.g., Austria: ÖKS15, Switzerland: CH2018). As such, the study is certainly relevant for the journal's readership. Its setup is sound and convincing, the results are presented in an appropriate manner and the conclusions are well-based on the results obtained. There are no language issues except for the mixed use of past and present tense in the presentation of the results, which should be revised. There are a few minor issues that should be corrected for as well as two major remarks (see below). However, I'd leave it up to the authors to consider these major comments or not. I believe a consideration would further improve the quality of the paper, but the study is sound and convincing even in its current state. My recommendation is therefore to return the paper to the authors for minor revisions.*

*Congratulations for this nice piece of work! Sven Kotlarski*

*MAJOR ISSUES*

*Cross validation: Similarly to the point raised by a previous reviewer, I believe that a proper cross validation framework would be helpful. MBCn is a more complex method than QDM, and there's an increased danger of overfitting. As MBCn explicitly corrects for biased inter-variable dependencies, snowfall amounts (if derived by a fixed temperature threshold) are well represented by definition. Being aware of the criticism by Maraun & Widmann, cross validation still makes sense in a split sample framework, e.g. by separating the historical period into the 15 coldest/warmest/driest/wettest years and the 15 warmest/coldest/wettest/driest years and using these subsets for calibration and verification, respectively. In case this splitting cannot be handled by HBV-light because the transient*

*character is lost, one could carry out a cross validation for at least one ERA-Interim EURO-CORDEX experiment (these experiments are available as well and are in basic temporal correspondence with the observations). In general a cross validation would make the point stronger that a multivariate BC is superior for the example presented.*

Yes, indeed a split sample would not be suitable input to a hydrological model. We can follow the reasoning of the referee, yet, as there have been a few new studies now on the pure bias correction, we would like to keep focus on the hydrological application here. In general, a lot of aspects have to be considered in the selection and application of a bias correction method for a given purpose. With our case study we mainly want to point to the potentially significant consequences in terms of snowfall fraction from a hydrological (modeller's) perspective, as also acknowledged by the referee above. Therefore we refer to our response to referee #1 regarding cross-validation (please see therein). For this study with a bivariate application case of MBCn, we consider it sufficient to briefly address this in the revision and refer again to past cross-validation evaluation efforts presented in the original paper about the MBCn method (Cannon, 2018).

*Reason for underestimated snowfall amounts by QDM: I understand that the paper puts an emphasis on the hydrological consequences of the two different BC methods. These effects are very well and convincingly presented. However, the question WHY QDM shows these deficiencies is not ultimately answered. The reason is to be found in the T-P relationship of the QDM data, and probably already appears in the raw RCM data. To analyze this further, 2D histograms would be extremely helpful and also illustrative.*

Below we add as an example a graph (for only one raw RCM data set, one RCP, one catchment) that shows distributions and bivariate probability density plots of P and $T_a$ in order to compare our HOCD, uncorrected, QDM-corrected, and MBCn-corrected data. Differences (biases) between the historical reference data (HOCD) and the uncorrected RCM data are evident. However, differences regarding the $T_a$–P intervariable relationship between QDM- and MBCn-corrected data are present (see e.g. local regression line in plots e–h) but more difficult to recognise in such kinds of plots. Hence, we included only the corresponding precipitation sums for days below and above 0 °C as shown in Figure 2 in the submitted version of the manuscript. However, figures such as the one below might be added (as supplementary material) to the revised version, if considered helpful. Furthermore, we agree that it is of high interest to discuss and understand the causes of the $T_a$–P intervariable-dependence-bias resulting in differences in temperature-threshold defined snowfall fractions better. We will extend the discussion a bit in this respect, but an ultimate answer may be beyond the scope of this study and would require separate investigations also based on an intercomparison for further observational datasets.

[Figure]

**Figure A: Exemplary representation of T$_a$ and P over the historical reference period 1977–2006 for the Schwarze Lütschine catchment according to the historically observed climate data (column HOCD) and uncorrected (column no BC), univariate-corrected (column QDM), and bivariate-corrected (column MBCn) climate model data from one GCM–RCM combination (ECEARTH–RACMO22E) for RCP 8.5. Top and bottom panel show marginal distributions of T$_a$ and P, respectively; centre panels show bivariate plots for T$_a$ and P with local regression lines (plots e–h) and density allocation (plots i–l).**

*MINOR ISSUES*

*Introduction and conclusions: The literature review should account for the studies by Wilcke et al. (Climatic Change, 2013) and Ivanov & Kotlarski (Int. J. Climatol., 2017). Inter-variable dependencies in standard QM have already been analysed in there. One of the results was that QM does not distort inter-variable dependencies as long as they are approximately represented by the raw RCM data. The results of the present work therefore indicate some distorted inter-dependencies already in the RCM raw output (which could be better described if my major comment #2 would be considered). These issues should also be discussed in the discussion/conclusions.*

Thank you, we are aware of these studies and will integrate references to them in the revised version and extend the discussion. See also comment above.

*p2 l4: The CORDEX data are actually not available from the CH2018 archive. The respective website only explains the selection of EURO-CORDEX models for the CH2018 Swiss climate scenarios. In the present study, EURO-CORDEX data were probably obtained from the ESGF archive.*

Thank you for this remark. We agree that the CH2018 archive is not the most appropriate reference. We will now reference the ESGF archive in relation to the download of the EURO-CORDEX data. In addition, we will expand our acknowledgements section to include the efforts of Dr. Urs Beyerle and to follow CORDEX terms of use (see point c):

https://www.hzg.de/imperia/md/assets/clm/cordex_terms_of_use.pdf

*Table 2: Just a note: The two runs driven by CNRM-CM5 are critical as the driving GCM CNRM-CM5 has an inconsistency in the historical period. It is fine to use them for the present work, but in future works they might have to be removed. More information is available from the new EURO-CORDEX errata page available from www.eurocordex.net.*

Thank you for pointing this out. We acknowledge the limitations of some of the individual EURO-CORDEX runs, which were pointed out to us during the CH2018 model selection process.

*Chapter 3.1: The description of the QM methods is incomplete in the sense that it is not clear if the correction has been carried out for the bulk series (all days independent of the time of year) or depending on the time of year (e.g., seasonal or DOY dependence). This information is critical, as a bulk correction could be responsible for the deficiencies of QDM in my opinion. I believe the authors employed a seasonally dependent BC, but this needs to be better explained (even if reference to Cannon et al. is provided).*

We agree that this needs to be more precisely explained in the revised version. Yes, we applied bias correction in a seasonally dependent fashion. Specifically, bias corrections were applied over 3 x 10 = 30-year sliding windows. This involved replacing the central 10-years and sliding forward 10-years for each 30-yr window, until the end of the projection period is reached. Within each window – to ensure an unbiased seasonal cycle – bias corrections were applied separately for each calendar month.

*p11 l30-31. Do you have any explanation for the higher mean streamflow amounts for QDM? Are differences in ETP involved?*

Potential evapotranspiration (ET) was kept the same for all model runs but actual ET simulations can vary depending on water availability and presence of snow cover. However, this is not a main driver for the observed differences in total streamflow. The slightly higher mean streamflow for QDM compared to MBCn is mostly the case for the Hinterrhein catchment and might be partly explained by one HBV-light model parameter, the so-called snowfall correction factor that can potentially tackle snowfall undercatch measurement errors (by parameter values > 1.0) as well as snow sublimation losses (by parameter values < 1.0). For the Hinterrhein catchment calibration of this parameter resulted in a value of 0.81, meaning that the model reduces any snow input by 19%. Since snow makes up the largest fraction of precipitation input in the alpine study catchments this snow-specific reduction due to the calibration parameter might finally result in a lower simulated streamflow for the MBCn-based data, for which the snow fraction is higher. In addition, slightly higher ice melt runoff simulations contribute to the higher mean streamflow amounts for QDM compared to MBCn over the historical reference period in case of the Schwarze Lütschine catchment.

Please note: to address the request of referee #1 meanwhile we have already repeated the calibrations of the hydrological model with changed settings (slightly shortened calibration period to allow for a validation period and SWE calibration for extended elevation range) and carried out all model runs based on the results of this new (improved) calibrations. We will explain this in detail and update all result figures based on this re-calibration of the hydrological model in the revised version. In reference to the reviewer's comment above, please find below the new results for Figure 5. The additional calibration efforts led to overall less pronounced differences in terms of total streamflow and also a better agreement with observed streamflow, while the systematic differences in the snow and rain component fractions are not affected by the changed parametrisation of the hydrological model.

[Figure]

**Figure B: As Figure 5 of the submitted manuscript version (see there for explanations) but showing results based on changed parametrisation after re-calibration of the hydrological model.**

We thank the referee for noting the following minor issues, which will be corrected in the revised version:

*p2 l20: "... which correct for biases in the data's entire distribution..."*

*p6 l20-21: "... until the multivariate distributions of bias-corrected and observed data match."*

*p14 l24: "...disappearance vary by over a decade ..."*

*p15 l31: "...empirical-statistical bias correction methods ..."*

---

## Author Response (AR1)

**Authors' response to editor's and referees' comments on hess-2018-317**
"Effects of univariate and multivariate bias correction on hydrological impact projections in alpine catchments" by Judith Meyer et al.

Dear Editor, dear Referees,

Thank you again for the evaluation of our manuscript and the many helpful comments.

Please find below our point-by-point reply in blue to all editor and reviewer comments *(in black)* followed by the track-change version of the revised manuscript.

Best regards,
Irene Kohn, on behalf of all authors

*Editor comments*
*Both reviewers indicate that it is an interesting paper but requires some redesign. I agree with this assessment. For example, in the intro, the authors mention the objectives of the study but do not mention the scientific hypothesis on this this study is funded. In fact, this word is not even mentioned in the manuscript. It is also important to make clear why the state of the art leads to formulate such hypothesis. Otherwise a manuscript becomes a case study. This could be done based on the proposed objectives.*

We revised the Introduction section and clarified our objectives and the underlying hypothesis.

*Technically, I would like to see a proper comparison with existing methods such as that of Hempen et al.: Hempel, S., K. Frieler, L. Warszawski, J. Schewe, and F. Piontek (2013), A trend-preserving bias correction: the ISI-MIP approach, Earth Syst. Dynam., 4(2), 219–236, doi:10.5194/esd-4-219-2013.*

In the submitted version of the manuscript we looked mostly for comparable studies that investigated consequences of a bi- or multivariate bias correction in hydrological impact modelling. We also agree with the suggestion of referee #2 that we should broaden our scope by considering the findings of other references that discussed inter-variable aspects in (univariate) bias correction methods also outside a hydrological modelling context. Consequently, we added such references in the Intro and Discussion Section. However, there are many more publications of further bias correction methods similar to the one mentioned above and also more recent ones. We don't agree that an even more comprehensive review of existing bias correction methods in general needs to be part of this manuscript, in which we exemplarily compared hydrological impacts of a uni- and a bivariate bias correction method in two alpine catchments. A more comprehensive discussion on (recent) bias correction approaches, including among others reference to Hempel et al. 2013, is presented in the original publications on the bias correction methods applied by us: Cannon et al. (2015) and Cannon (2018). The issue of preserving projected trends, being the main focus of Hempel et al., 2013, is treated in more depth in Cannon et al. (2015). However, **a reference to Hempel et al. (2013) was added in Section 3.1**.

*As mentioned by the both reviewers, a proper model validation and crossvaligdation should be carried out. Nowhere in the paper a validation period is mentioned. Model performances against observation are not presented in any form. This is a must in this kind of studies.*

Calibration of the hydrological model was carried out again with a slightly shortened period to allow for a short validation period. An additional table summarising model performance measures for the calibration and validation period is included in the revised manuscript and graphs showing model output vs. observation based data on streamflow, snow cover, and glaciers were added as supplementary material.

***Comments Referee 1***
*The manuscript "Effects of univariate and multivariate bias correction on hydrological impact projections in alpine catchments" by Meyer et al. investigates the effect of univariate versus (BC) multivariate bias correction (MBC) on the representation of snow, ice and rain representation in a climate change impact assessment approach. MBC has the advantage to control for variable interdependency (T and P namely) that in turn influences rain-snowfall fractioning. They used BC and MBC to bias correct and downscale 10 GCM-RCM combination using quantile mapping and drove the hydrological model HBV-light in a transient setting (1976-2099). They analyzed the effect of the different methods for snow-water-equivalent, icemelt, streamflow amounts and its composition (ice, snow, rain) over time. The paper adds to the ongoing discussion on the possible effects of intervariable dependency, especially by adding the information about the effect of streamflow composition. The paper is scientifically interesting, original, overall very well writing and certainly within the scope of the journal and of interest to the readers. Texts and figures are widely clear and lead the way for reasonable conclusions.*

*Beside some minor comments that I will list below, I have two more substantial concerns that refer to the design of the study.*

In summary, as we understand, these more substantial concerns are: a) hydrological modelling results for a validation period are missing, and b) the use of a constant temperature lapse rate in HBV. We provide detailed explanations and suggestions for compromises below. But summarized in brief, these decisions need to be seen in context with the aim of the study to test principal effects of uni- vs. multi-variate bias correction in such an environment. The objective of the study is not to present a best possible hydrological impact assessment for the sake of a case study (in line with the Editor's comment, this would not be a suitable study for HESS) and results for the two catchments should not be taken as that, i.e. interpreted quantitatively as future reality. Instead, the case study catchments (rather than e.g. a completely hypothetical catchment, which would be an alternative) allow demonstrating and discussing the potential effects; i.e. the objective is to compare simulation results for the same boundary conditions in terms of hydrological modelling only differing in the $T_a$ and P input series corrected with a uni- vs. a multivariate method. In the revised Introduction section we stated this more clearly.

The use of the set lapse rate is motivated by the small scale of the catchments and the fact that this is the commonly used approach in HBV applications. We argue that both do not affect our conclusions regarding the specific aims of this study, in which focus is on the comparison of systematic effects of bias correction methods, to keep the paper concise, and modelling with a conceptual model that is as parsimonious as possible. Note that Figure 2 (P at T > or ≤ 0 °C) compares results of QDM vs. MBCn independent of the hydrological modelling and related assumptions. Subsequent results of hydrological modelling are all reasonable consequences of that. Changes in the hydrological model parametrization or incorporation of changed lapse rates may have led to slightly changed simulation results but we are convinced that that would not lead to different main conclusions in the comparison of the simulations driven by univariate and multivariate corrected *P*- and *T*-input.

*1. The hydrological model was calibrated for the entire reference time period, e.g. 1976–2006, (page 7, line 16 ff) against streamflow, snow (SWE and snow-covered area), and glacier volume.*

*Three things puzzles me here:*

*1.1 There is no validation period!*

*1.2 Model performances against observation are not presented at all, neither as statistical measure nor in the graphs. I am aware that only differences between input data sets are analyzed in this study, still the basic performance measures are needed to frame the results. E.g. if the representation snow melt is not well captured (what I do not assume here) but the streamflow is (hence snow melt insensitive) than the snow sensitivity to changes in the input data might also be underrated.*

The reason for not reserving a validation period was to make full use of available observation-based data, in particular on glacier change. As indicated by the referee, we think that details of model performance and hydrological model validation are not of highest priority to compare the results of univariate and bivariate bias correction of precipitation and temperature in hydrological modelling of alpine catchments in a general way, as we intend with our study. However, we fully agree that model validation is an important component of hydrological modelling studies in general and that some information on model performance should be added as general information for the reader.

1.1.Validation period: In our case the historical reference period 1976–2006, which was used for model calibration, was chosen because (i) the used HYRAS climate data product unfortunately covers only the period ≤ 2006, and (ii) model warmup started in the year 1973, the simulation in 1976 and needs to go until at least 2003 in order to capture the climatology and to make full use of reliable data for the initial conditions and development of the catchments' glaciers. We consider the use of observation-based glacier data for both, model initialization and calibration, crucial. Hence, we used an estimate of glacier volume for the year 2003 based on glacier area (by Paul et al. 2010) in model calibration. Consequently, a calibration period of at least 1973/1976 to 2003 has been a result of the available glacier data for the years of 1973 and 2003. Our model calibration strategy was developed within a multi-catchment modelling study (Stahl et al. 2017), for which a consistent calibration scheme and consistent reference datasets for 49 alpine catchments (including their glaciers) were needed. We are not aware of any additional glacier observation-based data covering all glacierized parts of the modelled catchments Hinterrhein and Schwarze Lütschine that could have been used additionally in this study.

We agree that validation normally should be done but it was not used here because the study rather focused on the relative performances of the bias correction approaches than on the absolute model performance. However, to address the requests for a validation period, **we repeated model calibration for a period until the year 2003 only, which allows at least three years (2004–2006) to be used for model validation. Consequently, Figures 3–6 were exchanged with results based on the hydrological model parametrization based on this new calibration and text referring to the figures was adjusted where needed.**

1.2 Model performance: Performance is usually far from perfect. But considering that we deal with high-elevation catchments, incorporate streamflow, snow characteristics, and glacier data in calibration we consider it an acceptable real-world representation for the purpose of the study. **We included quantitative results on model performance for the calibration and the new validation period 2004–2006 and validation period based on the components of the multi-criteria objective function as additional Table** (Table 3) in the revised manuscript. **Obs. vs. sim. result graphs were**

**added separately as supplementary material** in order not to complicate the previously existing results graphs in which the focus should be kept on the comparison of QDM- and MBCn- based results.

*1.3 this concern about validation and performance measures also extends to both QM approaches (was a cross-validation framework used? provision of verification statistics is needed) –> more emphasis should be laid on the presentation of validation and the introduction of a validation period/validation framework is required*

The BC methods are optimized to the best possible transformation. We think, a randomized cross-validation would not provide any targeted and usable information regarding our aims of testing systematic effects of QDM vs MBCn. Because the univariate and multivariate bias correction algorithms are applied in an asynchronous fashion to freely running climate simulations – adjusting the marginal/joint distributions – it is almost guaranteed that they will perform well in terms of cross-validated measures of distributional fit. This is by construction. See also the critique by Maraun & Widmann in HESSD: https://www.hydrol-earth-syst-sci-discuss.net/hess-2018-151/.

Cross-validation does make sense when performance, especially for aspects not explicitly adjusted, is measured in a setting where climate model simulations are synchronized with the real-world climate state, for example in climate prediction or perfect boundary condition (e.g., reanalysis-driven) setups. We note that such reanalysis-driven evaluations have been performed in Cannon (2018) for the two algorithms used in this study. This was done over a large continental domain for a more complicated multivariate index (fire weather), that combines, in a nonlinear fashion, the current and lagged effects of temperature, precipitation, wind, and humidity. **We discussed these issues and past validation efforts in an additional paragraph in the revised Discussion section**.

*2. If I understand correctly, the combination of climate model data and the hydrological model is as follows:*
*- The quantile mapping is performed between climate model output and the average catchment value for T and P*
*- This mean value is interpolated within the catchment by a lapse rate that is fixed for each day of the year (extracted from the reference period)*

*This approach might be needed in HBV-light, but is based on the assumption that the lapse rate is not changing over time and is independent of certain events. This is a very strong assumption that is disproven by numerous study showing e.g. elevation depended warming, for instance. I assume, and to my own experience, that the slope of the lapse rate is quite sensitive to SWE simulations. Hence, this strong assumption likely influences the robustness of your results. Furthermore, you cannot control for this in the calibration of the model, as you limited the evaluation of model performances of SWE and snow-covered area to 2000 – 2500 m asl (page 7, line 21), which is exactly the catchment mean elevation for which the lapse rate is of minor effect. To me the fixed lapse rate is a very critical points in the study and need to be solved. A possible workaround of this issue would be to perform a quantile mapping that is not based on catchment mean values, but for each grid cell of the HYRAS data set. This procedure is currently done for the new CH2018 climate scenarios by MeteoSwiss. Doing so, you can extract the lapse rate for each day separately and use this dynamic lapse rate. With this procedure you would not have to make this strong assumption of a static lapse rate and have much more reliable results.*

Firstly, for clarification: calibration in terms of snow covered area fraction was based on the entire catchment only the calibration of SWE was limited to the elevation range 2000–2500 m a.s.l. Phrasing of the sentences referring to the calibration of snow characteristics in the submitted version is not wrong but a bit misleading. When we re-calibrated the model for the revisions we used a larger elevation range for the SWE calibration (see also separate comment on SWE calibration below).

We agree that lapse rates and their potential future changes are important for hydrological modelling of alpine catchments in particular and that hydrological impact assessments should attempt to incorporate them adequately. However, this is not easy and a topic related to high uncertainties anyway. The use of a linear gradient is a key feature of HBV. That is already a – if you like, strong, – simplification as one of many common in hydrological modelling.

We considered the possible ways to derive bias corrected lapse rates from GCM–RCM output:

- Referee's suggestion to apply BC on a grid-to-grid base to HYRAS: this would in fact be an application of BC to a grid distinctly higher in resolution (1x1 km) that is, as we mentioned in the paper, also criticized widely and would call for an explicit downscaling step, i.e. it would require aggregation and re-gridding of HYRAS. Consequently, results would also be strongly influenced by the underlying background climatology of this HYRAS interpolation product (also available only for recent conditions). We are not convinced that this is an appropriate/better solution.

- Theoretically, of course it is feasible to extract lapse rates from the GCM–RCM output and incorporate in the bias correction, including in case of the multivariate MBCn to apply it to 4 variables, not only to T and P catchment mean but additionally to lapse rates. Uncertainties in the extraction of lapse rates from 12x12 km CORDEX grid for mesoscale catchments (54 km² the smaller one) and their subsequent bias correction are expected to be substantial (even higher than those in case of the catchment averages), too. In addition, we think that evidence of trends in projected temperature lapse rates at the catchment scale is limited.

➔ Considering our aims and setting (model, data, objectives) **we don't think that extra efforts vs. expected added value at the expense of more assumptions may improve the confidence in the conclusion of the study**. We think it would rather complicate the interpretation of the results in terms of effects of the QDM vs. MBCn on $T_a$ and P. We prefer simple, comparable boundary conditions for our modelling experiment, and kept lapse rates unchanged/unaffected. As stated above, we stressed more clearly that our study should be taken as that, a modelling experiment (and not a perfect impact assessment). **Taking the long-term seasonal average lapse rate pattern from observed data as stationary was one of our assumptions to facilitate the separation of the effect of incorporating or ignoring interdependencies of catchment averages of $T_a$ and P in bias correction in the interpretation of results**.

*I think that these proposed changes are accomplishable in a reasonable time. Therefore, if those and the following comments are addressed, I am happy to comment on the manuscript again and likely recommend a publication. I am looking forward to the revised version.*

To summarise briefly: we re-calibrated for 1976–2003 with validation for 2004–2006, added information on performance, and we reconsidered the SWE calibration, i.e. extended it to a larger elevation range. While not crucial for our study's main objective, those revisions were straightforward

and time efforts justifiable. But from our side and with the study's objective in mind we do not see the need for cross-validation of bias correction and argue that fixed lapse rates can be used (see above).

*Specific comments:*

*Page 1, line 23ff. I suggest to state this result ("for the historical..:") prior to the effect on the future as this ensures an improved bias correction for MBC.*

Thanks for this suggestion. We rephrased those sentences slightly.

*Page 2, line 24: This publication might also be of interest (I am not an author): Wilcke,Renate Anna Irma; Mendlik, Thomas; Gobiet, Andreas (2013): Multi-variable error correction of regional climate models. In: Climatic Change 120 (4), S. 871–887. DOI:10.1007/s10584-013-0845-x.*

Thanks for this suggestion. As also suggested by referee #2 we added references to this study in the Intro and Discussion sections.

*Page 4 line 2: Only the "Unterer Grindelwald"-glacier is big ( ~ 6.biggest in Switzerland). It is a glaciated catchment, but covered by smaller glaciers. Please, rephrase. Therefor,also the following sentence needs to be rephrased.*

We agree, thanks. The sentences were corrected.

*Page 5, line 7ff: Please highlight that you apply catchment averages and that these averages are the "Target" in the quantile mapping approach (If I understood you correctly)*

Yes, we work generally with series of catchment averages also as "target" data as stated already at the beginning of the data section Page 4 line 16: "The resulting time series of catchment mean precipitation and temperature were used as input for the calibration of the glacio-hydrological model and as historically observed climate data (HOCD) for the bias correction." **We tried to stress this even more clearly in the revised manuscript by adding 'catchment mean' explicitly at several places, where it may be important to remind the reader on that**.

*Page 5, line 10: This is a very unusual time period as it crosses to climate normal periods. Do you have any reason of this time window. It hampers comparability to other climate change impact assessment studies.*

As already explained above, **the use of the period 1976–2006 was motivated by the modelling of glacier change**. Glacier data for the catchments (all glacierized parts) were available for the years 1973 (glacier area based on the inventory by Müller et al. 1976 / Maisch et al. (2000) and modelled ice thickness data provided by Matthias Huss) and 2003 (inventory by Paul et al. 2010). In our opinion, the initialization of the glaciers represents a major source of uncertainty for modelling glacierized catchments. Hence, we wanted to start the simulations (warmup) at a year for which glacier- area and -volume distributions are relatively well constrained by observation based data. Furthermore, we think it is important to incorporate glacier data in model calibration, what we realized by using glacier volume estimates based on glacier area data for the year 2003 (see also comment above). The used

HYRAS climate data product was updated recently but had not been available for periods beyond 2006 at the time of the study. Moreover, we would like to stress again that our study should be mainly considered as a general test to reveal and discuss principal "effects of univariate and multivariate bias correction" using two alpine catchments for demonstration rather than a specific climate impact assessment for the two case study catchments. We think, we clearly focused our interpretation of results on the comparison of univariate and bivariate bias correction; it is only limited possible to discuss the presented simulation results quantitatively in terms of projected impacts and to compare them with more comprehensive climate impact assessments.

*Page 5, line 13. Which gauging station was used. I am only aware of the FOEN station in Lütschine-Gsteig, and the Weisse Lütschine, Zweilütschinen. Did you use differences of these stations?*

Yes, exactly we used the records from the stations Lütschine–Gsteig and Weisse Lütschinen–Zweilütschinen to reconstruct streamflow 1973–2006 for the outlet of the study catchment (in mm/day) at the location of the station Schwarze Lütschine – Gündlischwand operated by the Canton of Bern. Observed stream flow at the station Schwarze Lütschine – Gündlischwand that was available for the period 1992–1999 was used to validate the reconstructed streamflow time series. **We explained this in the revised version**.

*Page 6, line 6: This is phrased wrongly. Univariate QDM cannot be both widely accepted (and used since several years) and developed by Cannon et al. 2015.*

We agree that this was not precisely phrased and revised this paragraph.

*Page 6, line 9: detrending of a time series is problematic, as assumptions about the kind of trend are necessary. Can you please add information about the way the trend is treated and comment on possible effects.*

The approach used for QDM follows the quantile delta change (Olsson et al. 2009, Atmos. Res.) and quantile perturbation (Willems and Vrac 2011, J. Hydr.) methods and is explained and discussed in detail in Cannon et al. (2015). Strictly speaking, the use of "detrending" in the submitted version was inaccurate. The detrended quantile mapping (DQM) algorithm in Cannon et al. (2015) *does* use a linear detrending step, but this is not the case with the QDM (and, by extension, MBCn methods). QDM adjusts all quantiles in the projection period by first removing the projected change signal (e.g., if the climate model projects a 20% increase in the $90^{th}$ percentile of precipitation in the future period, this change is removed, etc.), then it applies standard quantile mapping, and finally the projected change signal is reintroduced. **We revised Section 3.1 to be more precise with respect to the preservation of the climate models' change signal and added reference to Hempel et al. (2013) as suggested by the editor.** We do not think that any further detailed explanations and discussion are needed within the scope of this study, given that this is included in the original paper on the method

*Page 6, chapter 3.1: additional information about the validation procedure should be given as well as information about the "target value" (catchment averages).* Please, see our general comment on cross-validation above. The last sentence of this section already referred to the catchment averages being subject to bias correction. This sentence was slightly rephrased to place a bit more emphasis on that.

*Page 6, line 28: Please, quantify the difference by adding grid cell size and range of catchment area.*

Information on catchment sizes (54 km² and 180 km²) and the resolution of the climate model data (0.11°: ~12x12km) and the HYRAS data (1x1km) used to derive the catchment mean values are given in the corresponding subsections but were now repeated here (in brackets) and the sentence slightly rephrased.

*Page 7, line 16: so no separation of calibration and validation time period ! see general comments*

Please, see response to general comment above, too. The **original period was split to obtain a model validation, result graphs updated, and information on model performance added**.

*Page 7, line 21-22: is it correct that you not only evaluate but calibrate your model against this elevation limitation? Please rephrase.* See below for a combined reply to this and the next comment.

*Page 7, line 21-22: I disagree with the statement that only the area 2000-2500 is crucial as in my experience it is also very important for streamflow how much of the entire catchment is covered by snow – and hence contribute to snow melt. Please, comment.*

See also response to general comment above. For model calibration, originally, we used: a) snow covered area fraction of the entire catchment!, b) mean SWE for the elevation range 2000–2500 m a.s.l. Both were derived from a gridded SWE climatology product from the Swiss SLF institute. Due to discrepancies in the data product's resolution and the high detail of elevation-and-aspect zones for glacierized and non-glacierized catchment part for the catchment model, we think a comparison of mean values calculated for the entire catchment or a (not too limited) elevation range most appropriate. The reference data product from the SLF is known to be based on relatively few stations for elevations above 2500 m a.s.l. (in the domain of the study); thus, we constrained the use of the data to elevations below 2500 m a.s.l.. Selection of the elevation range of 2000–2500 m a.s.l. was based on an analysis of the SLF product for SWE statistics over all elevation zones of 49 alpine catchments (alpine catchments modelled in Stahl et al. 2017). Focussing on all those catchments, we considered the elevation range 2000–2500 m a.s.l. particularly important. However, **we agree that an extension of elevations in the catchments SWE calibration is useful for the catchments in this study. Hence, we extended the elevation range used for calibration of SWE to elevations ≤ 2500 m a.s.l. without a lower limit**.

*Page 8, Figure 2: What is also striking is that noBC is performing better than QDM for rainfall. Can you add on this?*

This is not the case. If understood correctly, your statement is "true" if one looks at the median of the boxplots for rain for days with $T_a > 0$ °C only (left part of left panel plots for historical period) ignoring the considerable bias for rain for days with $T_a \leq 0$ °C (right part of left panel plots for historical period). Looking at total precipitation independent of $T_a$, considerable biases towards too large precipitation sums are evident in the noBC data from all GCM–RCM combinations. Both methods (univariate QDM and MBCn) correct for this to a similar degree, with marginal distributions the same for both methods, by definition, as also stated in the manuscript. See also the additional

figures showing $T_a$ and $P$ distributions provided as supplementary material in response to the request by referee 2.

*Page 10, Figure 4: Maybe an error occurred, as the boxplots in the lower panel of the Schwarze Lütschine graph is missing.*

No, that is not an error but on purpose. As stated in the main text, for the Schwarze Lütschine catchments bias-corrected data from only few GCM–RCMs resulted in complete glacier retreat (volume = 0) by 2099. For those cases circles for the simulated year of complete glacier disappearance are plotted, whereas for most cases simulations resulted in a glacier volume ≥ 0 in 2099, thus no boxplot can be plotted. **We remarked on that in the figure caption**.

*Page 14, line 22: Please rephrase: It depends not on the bias correction but more specifically on the consideration of intervariable dependencies.*

Sentence was rephrased.

*Page 14, line 24: Are the found glacier retreat comparable to other findings?*

As stated above, we think one should discuss our simulations results with a focus on differences in the simulations (effects) as a result of the univariate vs. bivariate bias correction of temperature and precipitation rather than taking our results as specific future projections for the two catchments. We are not aware of projections specifically for the glaciers in our catchments that could be directly compared. Roughly, projections for mid-sized glaciers in the Aare basin might be compared with our simulations for the glaciers in the Schwarze Lütschine catchments and projections for small glaciers in the Alpine Rhine basin may be taken for comparison with our results for the glaciers in the Hinterrhein catchments. Results and discussion of projected glacier retreat and disappearance dates for small Swiss glaciers can be found in Huss & Fischer (2016, frontiers in earth science, https://doi.org/10.3389/feart.2016.00034); general results of projections for Swiss glaciers are presented, for instance, in the CCHydro report (FOEN, 2012: Effects of Climate Change on Water Resources and Waters. Synthesis report on "Climate Change and Hydrology in Switzerland" (CCHydro) project. Federal Office for the Environment, Bern. Umwelt-Wissen No 1217). In general, as far as a comparison is possible, the projected glacier retreat in our study compares well with results for projected glacier area change presented in those publications. However, note that i) it is generally difficult to compare the response and evolution of individual glaciers, ii) in our study **we consider the difference between the simulations driven by QDM- and MBCn-corrected data of much higher relevance than the exact date of glacier disappearance in comparison to other studies**.

*Page 15, line 8: Much more critical to me is the assumption of a fixed lapse rate, even more under climate change conditions*

Please, see general comments above. We agree that the question how to derive lapse rates for climate impact projections adequately is a difficult one and an important challenge requiring research efforts. However, **we restricted our study to demonstrating the effect of incorporating or ignoring interdependencies of catchment averages of $T_a$ and P in bias correction**.

*Comments Referee 2*
*The work by Meyer et al. presents an inter-comparison of a univariate and a multivariate bias correction (BC) method in terms of hydrological climate impact scenarios in two catchments of the Swiss Alps. For this purpose, daily temperature and precipitation amounts as simulated by ten EURO-CORDEX RCM experiments are bias-corrected toward observed catchment mean values and then fed into the HBV-light hydrological model. For BC the QDM and MBCn methods are employed, the latter taking explicitly into account variable interdependencies. The study finds important differences in the simulated streamflow for a historical period between QDM- and MBCn-based setups.*

*In general, shows MBCn shows a better performance. The main reason is an underestimation of snowfall amounts in the QDM-based setups (equivalent to a smaller snowfall fraction of total precipitation) which translates into smaller SWE amounts and an overestimation of winter streamflow while the spring meltwater peak is underestimated. The differences in the snowfall amounts between the two BC approaches furthermore translate into differences in future climate change signals of SWE, glacier coverage and, finally, streamflow. Qualitatively, the differences between the BC approaches are obtained for all ten climate model chains investigated, indicating a robust finding that seems to be valid for any GCM-RCM chain. In general, the paper is of very high quality and nicely highlights an important potential deficiency of bias-corrected climate scenarios in the Alpine region. It comes at a perfect time, as several recently released national reference scenarios are based on univariate BC approaches similar to QDM (e.g., Austria: ÖKS15, Switzerland: CH2018). As such, the study is certainly relevant for the journal's readership. Its setup is sound and convincing, the results are presented in an appropriate manner and the conclusions are well-based on the results obtained. There are no language issues except for the mixed use of past and present tense in the presentation of the results, which should be revised. There are a few minor issues that should be corrected for as well as two major remarks (see below). However, I'd leave it up to the authors to consider these major comments or not. I believe a consideration would further improve the quality of the paper, but the study is sound and convincing even in its current state. My recommendation is therefore to return the paper to the authors for minor revisions.*

*Congratulations for this nice piece of work! Sven Kotlarski*

*MAJOR ISSUES*

*Cross validation: Similarly to the point raised by a previous reviewer, I believe that a proper cross validation framework would be helpful. MBCn is a more complex method than QDM, and there's an increased danger of overfitting. As MBCn explicitly corrects for biased inter-variable dependencies, snowfall amounts (if derived by a fixed temperature threshold) are well represented by definition. Being aware of the criticism by Maraun & Widmann, cross validation still makes sense in a split sample framework, e.g. by separating the historical period into the 15 coldest/warmest/driest/wettest years and the 15 warmest/coldest/wettest/driest years and using these subsets for calibration and verification, respectively. In case this splitting cannot be handled by HBV-light because the transient character is lost, one could carry out a cross validation for at least one ERA-Interim EURO-CORDEX experiment (these experiments are available as well and are in basic temporal correspondence with the observations). In general a cross validation would make the point stronger that a multivariate BC is superior for the example presented.*

Yes, indeed a split sample test would be problematic as input to a hydrological model as there might be rather long memory effects. We can follow your reasoning, yet, as there have been a number of studies on the bias correction recently, **we would like to keep focus here on the hydrological application**. In general, a lot of aspects have to be considered in the selection and application of a bias

correction method for a given purpose. With our study we mainly want to point to the potentially significant consequences in terms of snowfall fraction from a hydrological (modeller's) perspective, as also acknowledged above. Therefore we refer to our response to referee #1 regarding cross-validation (above). **For this study with a bivariate application case of MBCn, we consider it sufficient to briefly address this in the revision and refer again to past cross-validation evaluation efforts presented in the original paper about the MBCn method (Cannon, 2018). Accordingly, a paragraph on these aspects was added in the Discussion Section.**

*Reason for underestimated snowfall amounts by QDM: I understand that the paper puts an emphasis on the hydrological consequences of the two different BC methods. These effects are very well and convincingly presented. However, the question WHY QDM shows these deficiencies is not ultimately answered. The reason is to be found in the T-P relationship of the QDM data, and probably already appears in the raw RCM data. To analyze this further, 2D histograms would be extremely helpful and also illustrative.*

Below we add as an example a graph (for only one raw RCM data set, one RCP, one catchment) that shows distributions and bivariate probability density plots of P and $T_a$ in order to compare our HOCD, uncorrected, QDM-corrected, and MBCn-corrected data. Differences (biases) between the historical reference data (HOCD) and the uncorrected RCM data are evident. However, differences regarding the $T_a$–P inter-variable relationship between QDM- and MBCn-corrected data are present (see e.g. local regression line in plots e–h) but more difficult to recognise in such kinds of plots. Hence, we included only the corresponding precipitation sums for days below and above 0 °C as shown in Figure 2 in the submitted and revised version of the manuscript. However, **figures as the one below but for all GCM–RCM combinations, both RCPs, and both catchments were added as supplement**. Furthermore, we agree that it is of high interest to discuss and understand the causes of the $T_a$–P intervariable-dependence-bias resulting in differences in temperature-threshold defined snowfall fractions better. **We extended the discussion a bit in this respect (first paragraph of Discussion section), but think an ultimate answer is beyond the scope of this study and would require separate investigations also based on an intercomparison for further observational datasets**.

[Figure]

**Figure A: Exemplary representation of T$_a$ and P over the historical reference period 1977–2006 for the Schwarze Lütschine catchment according to the historically observed climate data (column HOCD) and uncorrected (column no BC), univariate-corrected (column QDM), and bivariate-corrected (column MBCn) climate model data from one GCM–RCM combination (ECEARTH–RACMO22E) for RCP 8.5. Top and bottom panel show marginal distributions of T$_a$ and P, respectively; centre panels show bivariate plots for T$_a$ and P with local regression lines (plots e–h) and density allocation (plots i–l). See Supplement for figures for all applied climate model projection datasets and for both catchments.**

*MINOR ISSUES*

*Introduction and conclusions: The literature review should account for the studies by Wilcke et al. (Climatic Change, 2013) and Ivanov & Kotlarski (Int. J. Climatol., 2017). Inter-variable dependencies in standard QM have already been analysed in there. One of the results was that QM does not distort inter-variable dependencies as long as they are approximately represented by the raw RCM data. The results of the present work therefore indicate some distorted inter-dependencies already in the RCM raw output (which could be better described if my major comment #2 would be considered). These issues should also be discussed in the discussion/conclusions.*

Thank you, **references to those and further studies were added in the revised Intro and Discussion sections**. See also comment above.

*p2 l4: The CORDEX data are actually not available from the CH2018 archive. The respective website only explains the selection of EURO-CORDEX models for the CH2018 Swiss climate scenarios. In the present study, EURO-CORDEX data were probably obtained from the ESGF archive.*

Thank you for this remark. We agree that the CH2018 archive is not the most appropriate reference. **We now reference the ESGF archive** in relation to the download of the EURO-CORDEX data. In addition, we expanded our acknowledgements section to include the efforts of Dr. Urs Beyerle and to follow CORDEX terms of use (see point c):

https://www.hzg.de/imperia/md/assets/clm/cordex_terms_of_use.pdf

*Table 2: Just a note: The two runs driven by CNRM-CM5 are critical as the driving GCM CNRM-CM5 has an inconsistency in the historical period. It is fine to use them for the present work, but in future works they might have to be removed. More information is available from the new EURO-CORDEX errata page available from www.eurocordex.net.*

Thank you for pointing this out. We added a remark on that in the manuscript (footer, Table 2).

*Chapter 3.1: The description of the QM methods is incomplete in the sense that it is not clear if the correction has been carried out for the bulk series (all days independent of the time of year) or depending on the time of year (e.g., seasonal or DOY dependence). This information is critical, as a bulk correction could be responsible for the deficiencies of QDM in my opinion. I believe the authors employed a seasonally dependent BC, but this needs to be better explained (even if reference to Cannon et al. is provided).*

We agree with this critique. **The issue is now more precisely explained in the revised version of Section 3.1**. Yes, we applied bias correction in a seasonally dependent fashion. Specifically, bias corrections were applied over 3 x 10 = 30-year sliding windows. This involved replacing the central 10-years and sliding forward 10-years for each 30-yr window, until the end of the projection period is reached. Within each window – to ensure an unbiased seasonal cycle – bias corrections were applied separately for each calendar month.

*p11 l30-31. Do you have any explanation for the higher mean streamflow amounts for QDM? Are differences in ETP involved?*

Potential evapotranspiration (ET) was kept the same for all model runs but actual ET simulations can vary depending on water availability and presence of snow cover. However, this is not a main driver for the observed differences in total streamflow. The slightly higher mean streamflow for QDM compared to MBCn is mostly the case for the Hinterrhein catchment and might be partly explained by one HBV-light model parameter, the so-called snowfall correction factor that can potentially tackle snowfall undercatch measurement errors (by parameter values > 1.0) as well as snow sublimation losses (by parameter values < 1.0). For the Hinterrhein catchment the previous calibration (submitted manuscript version) of this parameter resulted in a value of 0.81, meaning that the model reduces any snow input by 19%. Since snow makes up the largest fraction of precipitation input in the alpine study catchments this snow-specific reduction due to the calibration parameter might finally result in a lower simulated streamflow for the MBCn-based data, for which the snow fraction is higher. In addition, slightly higher ice melt runoff simulations contribute to the higher mean streamflow amounts for QDM compared to MBCn over the historical reference period in case of the Schwarze Lütschine catchment. However, please note that **the additional calibration efforts (to address the requests by referee #1) led to overall less pronounced differences in terms of total streamflow and also a better agreement with observed streamflow, while the systematic differences in the snow and rain component fractions are not affected by the changed parametrization of the hydrological model (see Figure 5 in the revised manuscript)**.

Finally, we thank both referees for noting the following minor issues, which were corrected:

*p2 l20: "... which correct for biases in the data's entire distribution..."*

*p6 l20-21: "... until the multivariate distributions of bias-corrected and observed data match."*

*p14 l24: "...disappearance vary by over a decade ..."*

*p15 l31: "...empirical-statistical bias correction methods ..."*

*Page 6, line 22: "Climate model data" instead of data.*

*Page 15, line 34: is there a type? "re bias "*

[revised manuscript text omitted]